# Deviation from the matching law reflects an optimal strategy involving learning over multiple timescales

Kiyohito Iigaya [1,2,3,4,5], Yashar Ahmadian[1,6], Leo P. Sugrue [7,8], Greg S. Corrado[7,9], Yonatan Loewenstein[10], William T. Newsome[7] & Stefano Fusi [1,11,12]

Behavior deviating from our normative expectations often appears irrational. For example, even though behavior following the so-called matching law can maximize reward in a stationary foraging task, actual behavior commonly deviates from matching. Such behavioral deviations are interpreted as a failure of the subject; however, here we instead suggest that they reflect an adaptive strategy, suitable for uncertain, non-stationary environments. To prove it, we analyzed the behavior of primates that perform a dynamic foraging task. In such nonstationary environment, learning on both fast and slow timescales is beneficial: fast learning allows the animal to react to sudden changes, at the price of large fluctuations (variance) in the estimates of task relevant variables. Slow learning reduces the fluctuations but costs a bias that causes systematic behavioral deviations. Our behavioral analysis shows that the animals solved this bias-variance tradeoff by combining learning on both fast and slow timescales, suggesting that learning on multiple timescales can be a biologically plausible mechanism for optimizing decisions under uncertainty.

[1] Center for Theoretical Neuroscience, College of Physicians and Surgeons, Columbia University, New York, NY 10032, USA. [2] Gatsby Computational Neuroscience Unit, UCL, London W1T 4JG, UK. [3] Department of Physics, Columbia University, New York, NY 10027, USA. [4] Max Planck UCL Centre for Computational Psychiatry and Ageing Research, London WC1B 5EH, UK. [5] Division of Humanities and Social Sciences, California Institute of Technology, Pasadena, CA 91125, USA. [6] Institute of Neuroscience and Departments of Biology and Mathematics, University of Oregon, Eugene, OR 97403, USA. [7] Howard Hughes Medical Institute and Department of Neurobiology, Stanford University School of Medicine, Stanford, CA 94305, USA. [8] Neuroradiology Section, Department of Radiology and Biomedical Imaging, Universtiy of California, San Francisco, CA 94143, USA. [9] Google Inc., Mountain View, CA 94043, USA. [10] Department of Neurobiology, Edmond and Lily Safra Center for Brain Sciences, The Hebrew University of Jerusalem, 91904 Jerusalem, Israel. [11] Mortimer B. Zuckerman Mind Brain Behavior Institute, Columbia University, New York, NY 10027, United States. [12] Kavli Institute for Brain Sciences, Columbia University, New York, NY 10027, USA. Correspondence and requests for materials should be addressed to K.I. (email: kiigaya@gatsby.ucl.ac.uk) or to S.F. (email: sf2237@columbia.edu)

In a changing world, animals have to make decisions that are based on the limited information they can extract from the environment. Models of reinforcement learning and other learning theories assume that subjects' belief about the environment is gradually updated based on the history of choices and outcomes[1–5]. The optimal strategy for updating beliefs is to weigh past experiences according to the degree of uncertainty, or volatility, of the environment[6–8]. If the environment is stable, it is beneficial to consider a large number of past experiences to estimate the current state more accurately. In volatile environments, only a small number of recent experiences should be considered, as old ones may no longer be informative about the current situation.

Previous studies into flexible reward learning have largely focused on how animals can adjust their learning rate according to the volatility of the environment. Many of these studies have assumed that a single time constant (learning rate) is continuously modified over time (see e.g. refs. [2,6,7,9,10]). Real world environments, however, can change on multiple timescales and learning on multiple timescales can be highly beneficial computationally[8,11–17]. Here we analyzed an experiment to identify integration processes that operate on multiple timescales. We show that these processes cooperate to maximize the subjects' reward harvesting performance in a volatile environment. Interestingly, the subjects deviate from the behavior that would be optimal in a stationary environment. However, rather than a failure, this deviation reflects an adaptive strategy that is more efficient in volatile environments.

In the experiment, monkeys were trained to perform a dynamic foraging task[18] in which they track the changing reward rates of alternative choices through time. The reward rates changed periodically without any warning. Within each period during which the reward contingencies were kept constant, the probabilistic strategy that maximizes subjects' cumulative reward is to follow the so-called matching law, under a specific reward schedule that is often referred to as the concurrent variable-interval schedule[19,20].

According to the matching law, subjects distribute their choices across available options in the same proportion as the rewards obtained from those options[21,22]. This type of behavior has been observed across a wide range of species including pigeons, rats, monkeys, and humans[18,21–30]. Although the matching law provides a simple and elegant description of behavior, actual choice often deviates from matching.

For example, one common deviation, which is often referred to as undermatching, reveals itself as a more random choice allocation, because the subjects systematically choose less-rewarding options more often than the matching law predicts. Such deviations have been interpreted as a failure on the part of the subjects, reflecting poor discrimination between options[31], or noise in the neural mechanisms underlying decision making[32], or by an imbalance in the learning mechanisms[33]. In our data, we also observe significant undermatching; however, we find that overall harvesting performance improves as the behavior deviates more strongly from the matching law.

We hypothesized that this behavioral deviation, and accompanying performance improvement, may reflect an adaptation strategy with reward learning over multiple timescales, which optimizes the weighting of both recent and old experiences. Old experiences, when considered, should introduce a bias away from the current reward rates (which are unknown to the subject). This bias will instead reflect information about the long-run average of reward rates. For example, when different choices are equally rewarded in the long-run, the bias will be toward balanced choices, which would make the subjects choose the option with the lower reward rate more often than predicted by the matching

law if they knew the true rates. In our experiment, this bias translates into undermatching; therefore, choice behavior appears more random and exploratory than matching behavior, which would be optimal for a subject who knew the true reward contingencies.

However, the bias also has the effect of reducing fluctuations in the subject's estimate of current reward probabilities which must be inferred from finite, stochastic, observations. This reduction in the volatility of estimates compensates for any losses incurred by deviations from strict matching behavior, allowing the overall harvesting performance to increase when older experiences are taken into account.

To test this hypothesis, we estimated the time over which monkeys were integrating the rewards received for each choice. We found that this integration-time was much longer than has traditionally been described, varying slowly across days of experiments. As predicted, larger deviations from matching behavior correlated with longer integration-times, while longer integration-times corresponded to less volatile estimates of reward rates. We also found that the relative contributions of long timescale integrators correlated with the schedule of the experiments (e.g. with the duration of the time intervals between two consecutive experiments), suggesting dynamic adaptation over a long time period including times outside experimental sessions.

## Results

**The dynamic foraging task.** On each trial, the monkey is free to choose between two color targets by making saccadic movements (see Fig. 1a). Rewards are assigned to the two colors randomly, according to a concurrent variable-interval (VI) schedule, at rates that remain constant for a certain number of trials (block size: typically 100–200 trials). We call these experimentally controlled rates as the baiting rates. Once the reward is assigned to a target, the target is said to be baited, and the reward remains available until the target is chosen. This means that the probability of being rewarded for choosing a target increases with the time since the target was last chosen. In a stationary environment under this reward schedule, matching is known to be the probabilistic strategy that maximizes the average chance of obtaining rewards. In that sense matching could be considered optimal (see also ref. [34]). In this task the reward rates were not stationary, but were instead periodically changed in an unpredictable way. Nonetheless, the matching law still adequately captures the behavior of monkeys performing this task, as previously reported[18].

We plotted the fraction of times that monkeys choose one target versus the fraction of times that monkeys obtained a reward from the target in Fig. 1b. All datapoints are around the diagonal (blue). Notice, however, that there are clear deviations from the matching law, which become even more evident by comparing a linear fit (red line) of the datapoints to the diagonal. This is an example of the well-documented phenomenon of undermatching, whereby the choices of the animals appear to be closer to indifference (choice fraction close to 0.5) than would be predicted by the matching law.

We observed that deviation from the matching law varies over time (see different deviations estimated over two time intervals in Fig. 1b). One way to express this deviation more quantitatively is to compute the slope $S$ of the linear fit and compare it to the unitary slope of the diagonal. Therefore we will express the degree of undermatching as $1-S$. We found in data that this quantity varies significantly over time, ranging from 0.1 to 0.4.

The second observation is that changes in matching slopes are accompanied by changes in the overall performance that we can express as harvesting efficiency (i.e., the number of rewards that subjects actually obtained divided by the maximum number of

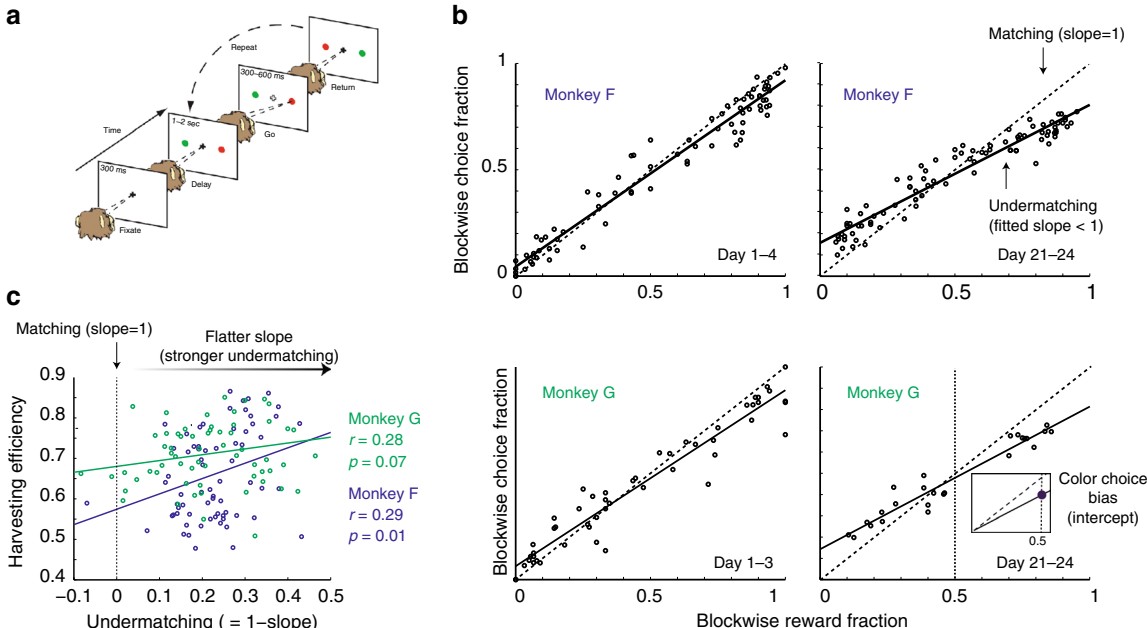

**Fig. 1** Task and behavior. **a** Behavioral protocol (adapted from[18]): the animal had to fixate the central cross, and after a short delay (Delay), it could make a saccadic eye movement toward one of the color targets (Go). If the chosen target was baited, a drop of water was delivered (Return). A delivery of reward resets the target to be empty, until it is baited again, which was stochastically determined with different baiting rates for different targets. The sum of baiting rates for two targets was set at ~0.35 rewards per trial. The relative baiting rates changed at the end of blocks (about every 100 trials) with no signal. The Ratio of baiting rates in each block was chosen unpredictably from the set (8:1, 6:1, 3:1, and 1:1). In this setup if the ratio is fixed, the matching law is known to approximate the optimal stochastic choice behavior. **b** Deviation from the matching law: the fraction of choices allocated to one target is plotted as a function of the fraction of rewards that were obtained from the same target for different experimental days (top left Monkey F days 1–4, bottom left: days 21–24, top right Monkey G days 1–3, bottom right: days 21–24). Each data point represents an estimate in a given block of trials, the solid line is a linear fit to the data. The matching law corresponds to a line with a slope equal to 1 (dashed line), while the observed behavior, with a slope <1, is called undermatching. Undermatching indicates that animals had a tendency to explore choices more (or, put simply, appear to be more random) than what the matching law would predict. For both monkeys the behavior deviates from the matching law, and the degree of undermatching (measured by the slope) changes over time. Note that undermatching is different from color choice bias, which is indicated by the filled circle in the inner panel (bottom right). The color choice bias is defined by the intercept of the fitted matching slope and the reward fraction of 0.5. **c** Paradoxically, the harvesting efficiency, which indicates how well the monkeys collected rewards, positively correlates with the degree of undermatching: the more choice behavior deviates from the matching law, the higher the harvesting efficiency. The harvesting efficiency is defined as the number of rewards that monkeys actually obtained divided by the maximum number of rewards that could have been collected. Hence it varies between 0 and 1. The monkeys almost always undermatched, the degree of which shows a wide distribution over sessions

rewards that they could have collected). Interestingly, as the subject's behavior deviates more from the matching law (see Fig. 1c), harvesting efficiency increases, an observation that might seem at odds with the optimality of matching behavior. However, as we will explain in the next section using computational models, this result makes sense in a non-stationary environment in which the reward probabilities change over time, and in fact signifies a learning strategy that involves both fast and slow parallel reward integration processes.

Note that there is another well-documented deviation from the matching law[31], which we refer to as color choice bias. This is a bias toward one of the colored targets, and can be quantified as the deviation of the linear fit to the behavior data from the unity diagonal predicted by the matching law at a reward fraction of 0.5 (see the inner panel of Fig. 1b and Methods). Undermatching (slope) and color choice bias (intercept) are independent behavioral measures. In the following theoretical analysis, we will primarily focus on undermatching, but we will come back to color choice bias later.

**Reward integration over multiple timescales leads to undermatching.** One common way to capture behavior in a dynamic foraging task is to build a model that integrates rewards over a certain number of trials. In a non-stationary situation in which

reward contingencies change from time to time, subjects need to adapt the timescale of reward integration to the volatility of reward schedules. When reward contingencies change rarely, it is better to integrate a large number of trials to improve the estimate of reward rates (from now on, we also refer to reward rates as choice values). Conversely, if reward contingencies change often, subjects should only rely on recent experiences, as more distant history does not reflect the current reward rates. One way to address this meta-learning problem for adjusting timescale is to run multiple, fast and slow, integrators in parallel, each of which integrates the reward streams from a specific target on a different timescale. The adaptation would then be to adjust the relative contributions of such integrators to choice.

The mechanism is described schematically in Fig. 2a. Consider the case of two exponential integrators characterized by two time constants $\tau_{Fast}$, $\tau_{Slow}$. There are two integrators (slow and fast) per choice, represented in the figure as boxes. The two top ones integrate the reward stream from the green target, whereas the two bottom ones integrate the reward stream from the other, red target. The outputs of these integrators approximate reward rates on a certain number of recent trials, which is determined by time constants, $\tau_{Fast}$ or $\tau_{Slow}$. We define the local income[18] of each target as a weighted average of the outputs of the fast and slow integrators for that target.

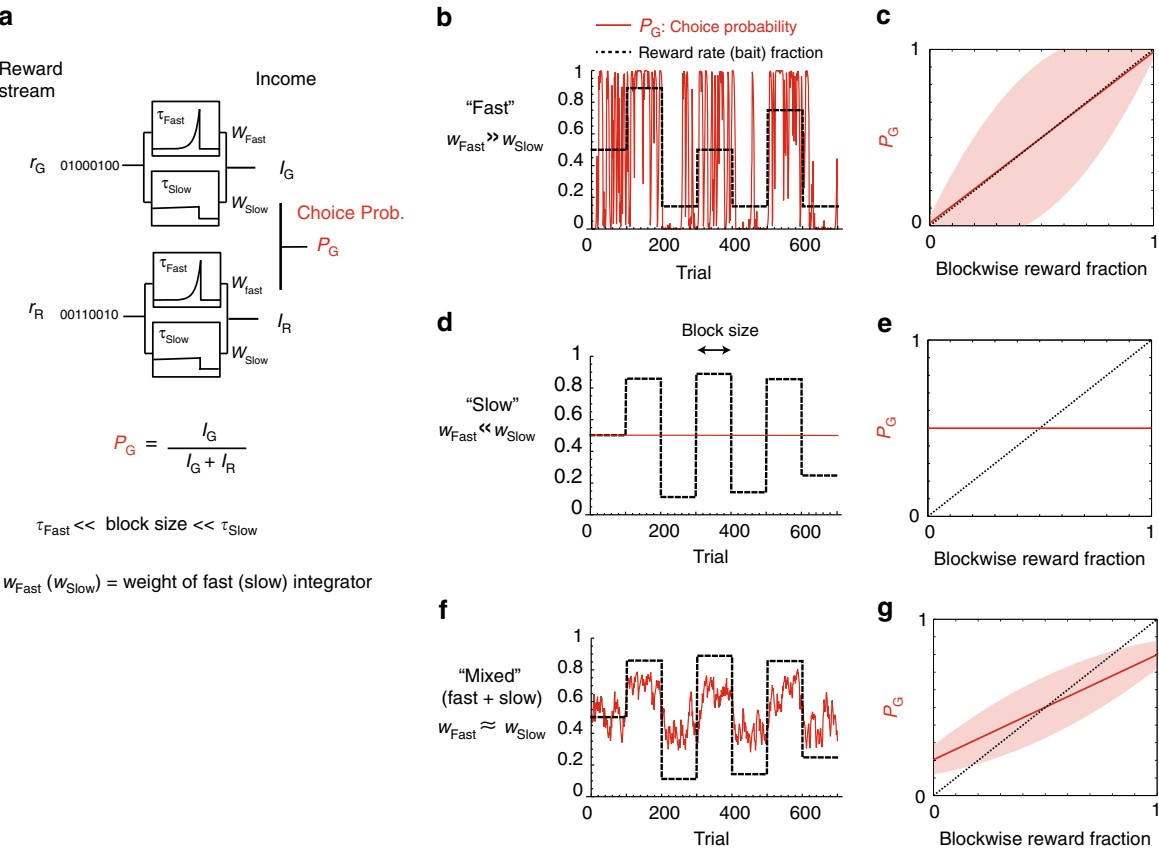

**Fig. 2** Model analysis: how slow reward integration leads to undermatching. **a** Scheme of a model. The model integrates reward history over two timescales ($\tau_{Fast}$, $\tau_{Slow}$) to estimate the expected income for each choice (red or green). These incomes are then combined to generate stochastic decisions, where $P_G$ ($P_R = 1 - P_G$) is the probability of choosing the green (red) target. While previous models[18,25] focused on the integration timescales that are shorter than the block size, here we assume that the long timescale $\tau_{Slow}$ is much longer than the block size, while the short timescale $\tau_{Fast}$ is still shorter than the block size. The model integrates the incomes estimated over the two timescales with adjustable relative weights: $w_{Fast}$ for the relative weight of the fast integrator ($\tau_{Fast}$) and $w_{Slow}$ for the weight of the slow one ($w_{Fast} + w_{Slow} = 1$). **b**, **c** Fast integration ($w_{Fast} \gg w_{Slow}$). If the weight of the fast integrator is much larger than the slow one, the model relies only on the recent reward history estimated over an interval that is approximately $\tau_{Fast}$. As a consequence, the estimated incomes largely and rapidly fluctuate. This noisy estimation leads to large fluctuations in the choice probability $P_G$ (red). Despite the fluctuations, the mean of such choice probability follows the matching law (indicated by the solid red line in **c**). However, the fluctuations of $P_G$ are rather large, as indicated by the broad shaded area, which denotes the standard deviation of $P_G$. **d**, **e** Slow integration ($w_{Fast} \ll w_{Slow}$). If the weight of the slow integrator is much larger than the fast one, the model now integrates rewards only on the long timescale $\tau_{Slow}$. This eliminates the fluctuations in the choice probability; however, the choice probability of is constant at 0.5, because the estimated incomes are balanced over multiple blocks of trials. Therefore, the choice probability becomes independent from the recent reward history, causing a strong (exploratory) deviation from the matching law (**e**). Note the the actual choice probability is determined by the overall color reward imbalance in the task (0.5, if no bias). **f**, **g** Mixed integration: ($w_{Fast} \simeq w_{Slow}$). If the two integrators are similarly weighted, both deviation from the matching law (undermatching) and the amplitude of the fluctuations are intermediate. This captures experimental data, and manifests a computational tradeoff between bias (long integrator; undermatching) and variance (short integrator; fluctuations). Parameters were set to be $\tau_{Fast} = 5$ trials, $\tau_{Slow} = 10,000$ trials, $w_{Slow} = 0.3$ for **f**, **g**. Note that our results do not rely on the precise choice of $\tau_{Fast}$ and $\tau_{Slow}$

The model's decision to choose a target is determined by a comparison between the local incomes for the two targets (see Methods for details). Following[18], choices are generated by the probability of choosing the green target given by:

$$P_G = \frac{I_G}{I_G + I_R}, \qquad (1)$$

where $I_{G/R}$ is the local income for Green/Red target. Although this is not the only way of modeling decisions that depend on past experiences, this has been shown to describe well the behavior[18,25]. In the previous analysis of Sugrue et al.[18] only a single relatively short timescale ($\tau_{Fast}$) has been considered.

The statistics of model's decisions change according to the relative contributions of fast and slow integrators. Consider the case in which $\tau_{Fast}$ is short (shorter than the typical block length

expressed in number of trials), and $\tau_{Slow}$ is very long (longer than the typical block length), so that the second integrator with $\tau_{Slow}$ integrates reward streams over multiple blocks of trials. If the weight $w_{Fast}$ of the fast integrator is much larger than that of the slow integrator $w_{Slow}$ (Fig. 2b, c), then the model rapidly tracks the recent average reward fraction. Fast learning is especially advantageous when adapting to rapid changes in reward contingencies (Fig. 2b). However, a disadvantage is that reward rate estimates fluctuate wildly (Fig. 2b). In Fig. 2c, we plotted the choice fraction vs the reward fraction, where the average of many blocks of trials (solid line) is very close to the diagonal, indicating that the model follows the (block-wise) matching law, but with a large variance (shaded area).

Conversely, if the weight of the slow integrator is so large that decisions are mostly driven by estimates on the long timescale

$\tau_{Slow}$, the local incomes become constant and approximately equal (Fig. 2d, e). Indeed incomes from two targets are approximately balanced on a long-run by experimental design. As a result, model's choice shows an extreme undermatching with negligible variance in choice probability (Fig. 2e).

Intermediate situations can be constructed by changing the relative weights of the two integrators (Fig. 2f, g). Increasing the weight of the slow integrator $w_{Slow}$ would increase deviation from the matching law, but it would also decrease variance in choice probability. This indicates that the model trades deviation from the matching law (undermatching) for a reduction in variance of inference about reward rates, by changing the relative contributions of fast and slow integrators.

**Bias-variance tradeoff in inference and behavior**. It is natural to ask whether there is an optimal value of the weight $w_{Slow}$. To address this question, we analyzed our model analytically in a more general situation in which we vary the degree of volatility (the block size) as a free parameter (see Supplementary Notes). For simplicity, we considered a task with one target in which subjects has to estimate dynamically changing reward rates

(income) over trials, or equivalently, estimate the bias of a coin, where the bias is fixed over each block of trials but it changes across blocks of trials. This simple inference task is closely related to our actual experimental task because in the experiment subjects also need to estimate reward rates from two targets accurately, and choose between targets following the local matching law Eq. (1). We will then confirm our analytical calculation results in simulations of the actual experimental schedule experienced by the monkeys.

In the analytical calculation, we find that the total error of the model's inference about reward rates can be generally expressed as a sum of two terms: one term expressing the difference between the model's average estimates and true reward rates, and the other term expressing the variance of model estimates, which is rooted in noisy, stochastic deliveries of actual reward. We refer to the former as to the bias of inference, and the latter as to the variance of inference.

Figure 3a–c show that there is a $w_{Slow}$ that maximizes the accuracy of inference about reward rates (i.e. it minimizes the mean squared error) and the value of $w_{Slow}$ depends on the volatility of the environment. In Fig. 3a, we contrast the squared bias of inference and the variance of inference as a function of

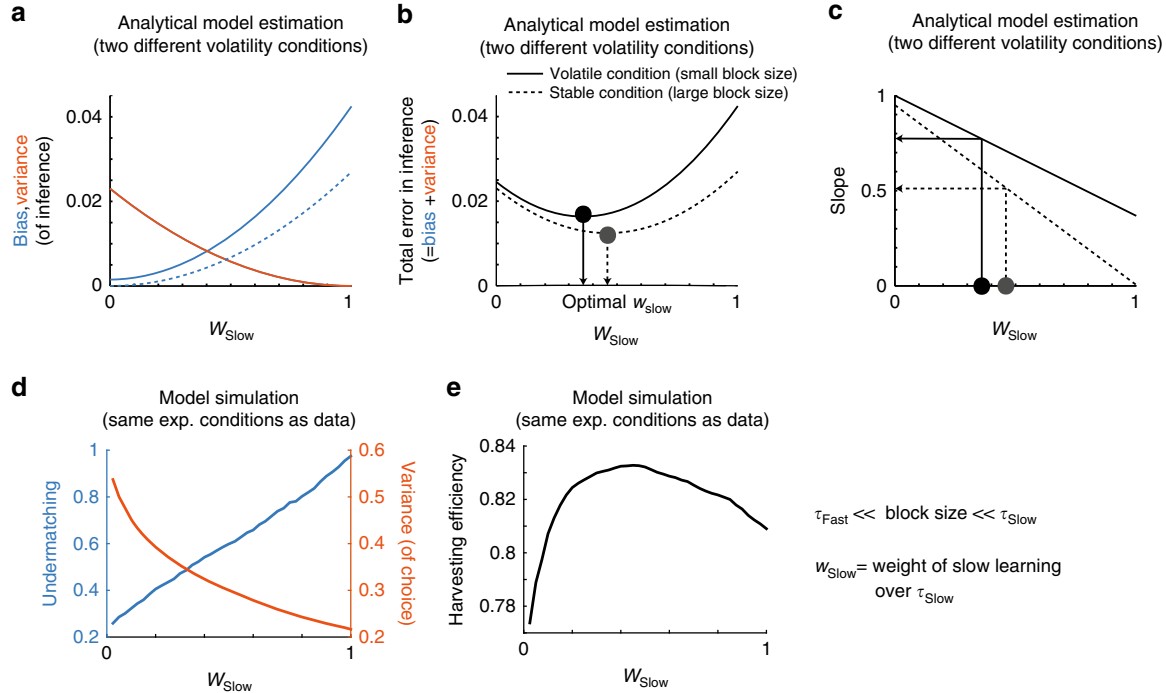

**Fig. 3** Model analysis: the bias-variance trade-off and the optimal choice behavior under uncertainty. **a–c** Analytical model results for two different volatility conditions, with a block size of 100 trials (more volatile) or 10,000 trials (less volatile). All the solid (dashed) lines refer to the results for a less (more) volatile task with a block size of 100 (10,000 trials). **a** Bias (blue) and variance (orange) of model's inference about reward rates show a tradeoff as a function of the relative weight of the slow integrator ($w_{Slow}$). The bias is squared. **b** The squared error of model's inference about reward rates, which is the sum of the squared bias (blue in **a**) and the variance (orange in **a**), is plotted against $w_{Slow}$ for different volatility conditions. The optimal relative weight $w_{Slow}$ for the volatile environment (solid vertical line) is smaller than that for the stable environment (dotted vertical line), since more volatile environments (solid curve) require faster integrators, or a smaller relative weight of the slow integrator. **c** Deviation from the matching law (shown as the slope of block-wise choice fraction vs reward fraction) covaries with the relative weight of the slow integrator, and also with the volatility condition. **d, e** Model simulation results on the same experimental schedule experienced by monkeys. **d** Model simulations show a clear tradeoff between undermatching (a form of behavioral bias) and the variance of choice probability, as a function of the relative weight of slower learning $w_{Slow}$. The square root of variance is shown for illustration. **e** Model simulations also show changes in harvesting efficiency as a function of the relative weight of the slow integrator $w_{Slow}$. As a result of the bias-variance trade-off, the curve takes an inverted U-shape, with a maximum at the optimal relative weight, determined by the volatility of the experiment. For panel **d**, we computed the variance of Monkey's choice probability as follows. First, monkey's choice time series was smoothed via two half-gaussian kernels with standard deviations of $\sigma = 8$ trials and $\sigma = 50$ trials, with a span of 200 trials. This gave us two time series: a fast one with $\sigma = 8$ trials and a slow one with $\sigma = 50$ trials. We defined the variance as the variance of the fast one over the slow one. The squared root of variance is shown in the panel. For panels **d, e** models with different $w_{Slow}$'s were simulated on the experimental schedules experienced by Monkey F. We set $\tau_{Fast} = 2$, $\tau_{Slow} = 1000$ trials. Note that our results do not rely on the precise choice of $\tau_{Fast}$ and $\tau_{Slow}$ (see also Fig. 1)

$w_{Slow}$. The variance of inference decreases but the bias of inference increases as the weight $w_{Slow}$ increases. The mean squared errors of model's inference, in turn, take an inverted U-shape (Fig. 2b). Not surprisingly, the optimal weight of the long timescale is larger for a more stable environment (dashed line), than for a more volatile environment (solid line). Consequently, the slope of matching behavior generated from the inferred reward rate changes according to the volatility (Fig. 3c).

We now show how our analytical model's predictions can be applied to the actual experimental task. We first note that the variance of inference is related to the variance of choice in our experiment. To see it, we simulated our model with a single learning rate (the same as in Sugrue et al.[18]). We used the parameters that were estimated by model fitting (maximum likelihood) for each session of the experiments for Monkey F. Supplementary Figure 2 clearly shows a positive correlation between the variance of inferred reward rate and the variance of choice behavior.

Therefore the bias-variance tradeoff in the inference, which we showed using analytical calculations, translates into an analogous tradeoff that should be observable in subject's choice behavior in the experiment. To see it, we simulated our model illustrated in Fig. 2a for different values of $w_{Slow}$ on the same experimental conditions experienced by Monkey F. In Fig. 3d we show that undermatching (a form of behavioral bias) is indeed traded off against variance in choice. A bias of 0 would indicate that the behavior follows the block-wise rates predicted by the matching law. As $w_{Slow}$ increases, undermatching $(1-$ the slope of matching behavior) becomes more evident. This, however, leads to a reduction in the variance of choice. Thus as seen in Fig. 3e, the

overall performance, measured by harvesting efficiency, has a maximum at an intermediate value of $w_{Slow}$.

Our computational model analysis suggests that the observed changes in matching behavior can be accounted for by changes in the relative contribution of a (extremely) slow reward integrator to decision making, and might reflect a tradeoff between bias and variance in value estimation. In this tradeoff, the relative weights of fast and slow integrators, $w_{Fast}$ and $w_{Slow}$, can be tuned according to the volatility of reward schedules.

**The predicted bias-variance tradeoff in the data.** We now test our theoretical predictions in the actual experimental data. The overall goal is to confirm the link between three features in data: the bias-variance tradeoff, undermatching behavior, and multi-timescale learning of reward history.

First, we provide experimental evidence for the bias-variance tradeoff. In the actual experimental data, we estimated bias (as undermatching) and the variance of choice, as well as the harvesting efficiency for each experimental session. As we estimated these measures without relying on model-fitting, we call this a model-independent analysis (see Methods for details). Figure 4a–d show that undermatching, variance, and harvesting efficiency varied dynamically over sessions. As we predicted from our model analysis, undermatching is negatively correlated with the variance (Fig. 4e). Analogously, the harvesting efficiency (Fig. 4c, d) is correlated both with the variance (negatively) and undermatching (positively) as shown in Fig. 4f, g respectively, suggesting that monkeys performed better when variance of choice were reduced and deviations from the matching law

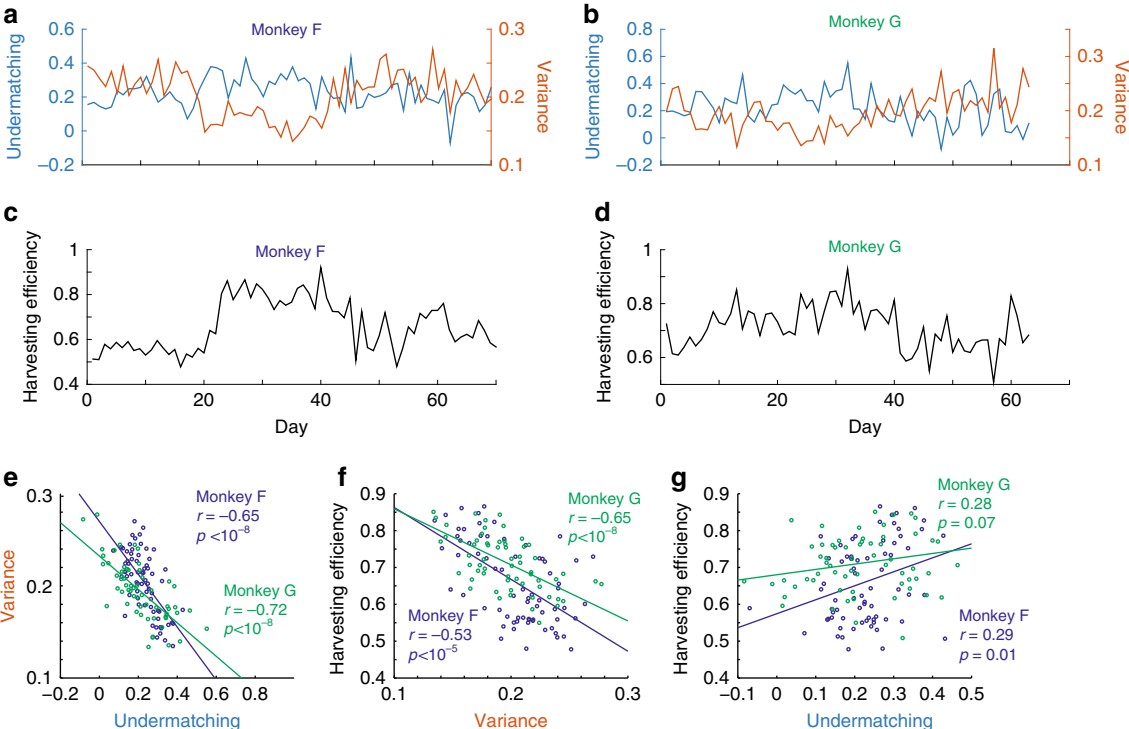

**Fig. 4** Monkeys show the predicted tradeoff (model-independent analysis). **a–d** Changes in undermatching, variance of choice and harvesting efficiency over experimental days. **e–g** Monkeys show the bias-variance tradeoff. **e** Undermatching $(1-$ the slope of matching behavior) and the variance of choice are significantly correlated negatively [permutation test: $p < 10^{-8}$ for Monkey F and $p < 10^{-8}$ for Monkey G]. This means that the monkeys trade off the bias (undermatching) against the variance of estimation, as predicted by our model. **f** By reducing variance of choice, monkeys improved their harvesting performance. The harvesting efficiency is significantly negatively correlated with the variance of choice [permutation test: $p < 10^{-8}$ for Monkey F and $p < 10^{-5}$ for Monkey G]. **g** Although increasing bias is harmful, benefits from reducing the variance of choice outweigh the costs [permutation test: $p < 0.07$ for Monkey F and $p < 0.01$ for Monkey G]. Note that in these figures squared root of variance is shown as variance

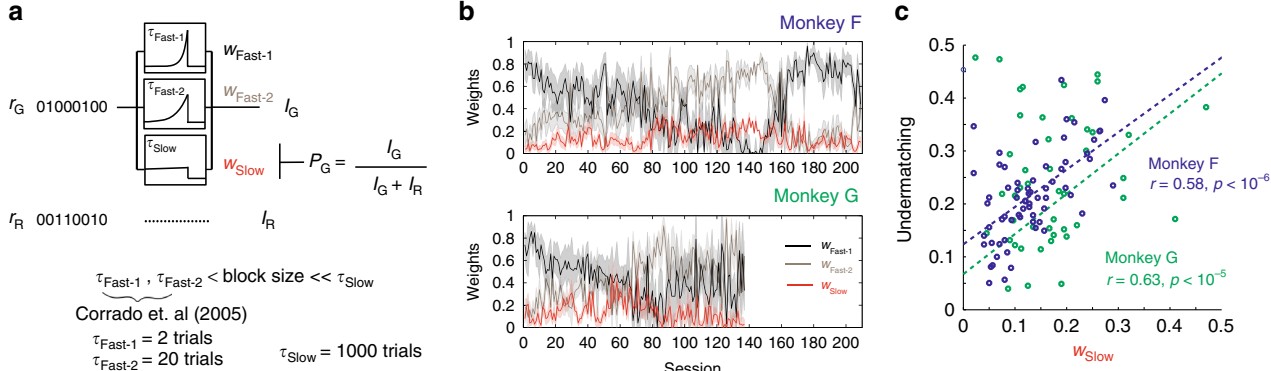

**Fig. 5** Fitting the multi-timescale model of Fig. 2 to the data. **a** A model with three timescales ($\tau_{\text{Fast}-1} = 2$ trials, $\tau_{\text{Fast}-2} = 20$ trials, and $\tau_{\text{Slow}} = 1000$ trials) is fitted to the data by tuning the weights $w_{\text{Fast}-1}$ (black), $w_{\text{Fast}-2}$ (brown), and $w_{\text{Slow}}$ (red) of the different integrators for each session independently ($w_{\text{Fast}-1} + w_{\text{Fast}-2} + w_{\text{Slow}} = 1$). **b** The weights of different timescales change over consecutive experimental sessions. The short timescales are dominant in early sessions, but the longer timescale becomes progressively more influential. The opposite trend is observed around session 160 of monkey F, probably due to the shortening of experimental sessions and longer inter-experimental-intervals (see Supplementary Figure 16 for more details). **c** Deviation from the matching law is correlated with the weight of the reward integration on a long timescale. Undermatching, computed over the last 50 trials of each block to ignore transients, is plotted against the fitted value of $w_{\text{Slow}}$, the weight of the longest reward integration timescale. Both monkeys show significant correlations between undermatching ($1 - $ slope) and $w_{\text{Slow}}$. We found, however, this model-based analysis inconclusive, as the correlation is expected from the model structure (please see the text)

became more prominent. This supports our hypothesis that monkeys used multi-timescale learning, particularly of a very long time constant, in order to optimize their choice.

**Undermatching and reward integration over long timescales**. If the bias-variance tradeoff is mediated by reward integration on long timescales and reward learning takes place over multiple time scales in parallel (as described in Fig. 2), then we predict that deviations from matching behavior should be strongly modulated by the relative contribution of the slow integrator.

An obvious way to test this prediction is to fit the multi-timescale learning model illustrated in Fig. 2a to the data, determine how the relative weights of the integrators change over time, and show that these changes correlate with changes in undermatching. We performed this analysis by using the model of Fig. 2 with three time constants instead of two. We included two time constants (~2 and 20 trials) that are shorter than the typical block size (as in Corrado et al.[25]), in addition to a third one that is significantly longer than the block size (1000 trials). The existance of such a long time constant is also supported by our auto-correlation analysis (Supplementary Figure 13). As we will explain below, the precise choice of the longest time constant does not affect our results.

We fit the model independently to the data from each session, by using maximum likelihood estimation of the relative weights of the three integrators ($w_{\text{Fast}-1}$, $w_{\text{Fast}-2}$, and $w_{\text{Slow}}$). We were surprised to find unusually smooth changes in these weights across sessions (Fig. 5b), as we did not impose any smoothness constraints to our fitting process (i.e. we fitted each session independently; for details see Methods). This suggests that the optimization of relative weights takes place slowly and continuously over sessions.

Further, as we predicted, changes in the weight of the slow integrator are correlated with changes in undermatching (Fig. 5c). However, due to a potential confound, this correlation is not sufficient to prove the link between undermatching and slow reward integration. This is because the slow reward integrators act as a bias that changes slowly over time, and any slowly changing bias will lead to correlations with undermatching, regardless of whether that bias is driven by slow reward

integration or some other process unrelated to reward history. For example, a bias that is randomly modified from session to session would also show correlations with undermatching, even though that bias does not depend on reward history. To better understand this confound, see Methods for a more detailed explanation.

As a direct test of this link between slow reward integration and undermatching, we estimated the effects of very slow reward learning in the experimental data, by measuring long-term reward-choice correlations across, rather than within, experimental sessions.

To directly measure the timescale of slow reward integration from the data (illustrated in Fig. 6a) we decided to exploit the other type of choice bias in matching behavior, which we refer to as color choice bias. A color choice bias in matching behavior is defined as the intercept, at reward fraction=0.5, of a line fit to block-wise matching behavior plotted in choice fraction vs. reward fraction (see inner panel in Fig. 1b). If animals indeed integrated rewards on a very long timescale, this long-term choice bias should be influenced by imbalance in past reward experience in which the experienced ratio of rewards received from each the two colors deviates from 50% (e.g. see ref. [16]). We can measure the slow integration timescale using session-by-session estimates of color imbalance in rewards and color bias in choice, by asking over how many sessions color reward imbalance influences future color choice bias.

To understand how this measure is related to the timescales of the integrators of Fig. 2, it is useful to run simulations of the model. We estimated the color reward imbalance and the color choice bias for each session of the simulated data. The reward color imbalance is defined as the fraction of reward obtained from one side minus 0.5, while as we previously defined, the color choice bias is the intercept of the matching slope at reward fraction=0.5. We then took the lagged causal correlations between these two bias vectors (Fig. 6b). Since the model learns reward history over trials, the color choice bias is supposed to be influenced by the color reward imbalance, with the correlation between the two quantities decaying as the lag increases. We then introduced what we refer to as the longest measurable integration timescale (LMIT), which is the longest time lag for which the correlation is significant. We estimate it by fitting a line to the

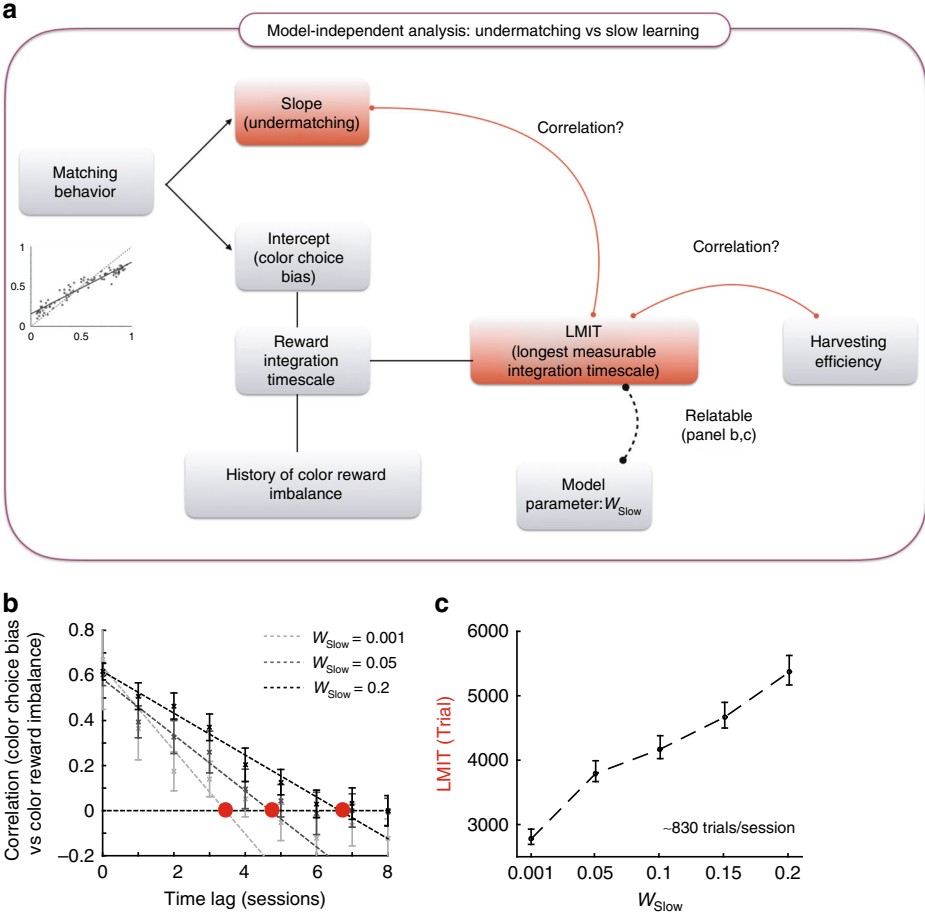

**Fig. 6** The model-independent analysis of integration over long timescales: the longest measurable integration timescale (LMIT). **a** The schematic of the analysis. From the plots describing matching behavior (see Fig. 1b), we fit a line to the datapoints and we measure its slope (undermatching) and its intercept at $x = 0.5$ (color choice bias). We also estimate session-by-session color reward imbalance that we defined as the imbalance between rewards obtained from two color targets. We then estimate how long ago color reward imbalance can influence future color choice bias by lagged correlation analysis, which we refer to as the longest measurable integration timescale (LMIT). Although this is a model-independent measure, it is related to our model parameter $w_{Slow}$. The motivation for introducing the LMIT is to show that the LMIT co-varies with the undermatching slope over time, providing direct evidence for the link between undermatching behavior and reward history integration over very long timescales. **b** Precise definition of the LMIT and its dependence on the relative weight $w_{Slow}$ of the slow integrator of the model (simulated data). The lagged correlation between color choice bias and color reward imbalance are estimated with data simulated using the model in Fig. 2a employing different relative weights of the slow integrator $w_{Slow}$. The correlation decays as the time-lag between the color choice bias and the color reward imbalance increases. The correlation points are fitted by a line using weighted linear regression (dotted lines), and the LMIT is defined by the point at which the fitted line crosses the zero correlation (red filled circles). The correlations are the mean of over 50 simulations and the error bars indicate the standard deviations. The model was simulated over the same reward schedule experienced by Monkey F. The timescales are assumed to be $\tau_{Fast} = 5$ trials and $\tau_{Slow} = 1000$ trials, though the precise choice of timescales is not essential for the results. **c** The LMIT estimated through simulations vs the weight of long-time constant in the model. The estimated LMIT shows a clear positive correlation with the relative weight of the long timescale $w_{Slow}$ employed in the model simulations. The LMIT is expressed in trials, by converting from sessions using the mean session size. This strong monotonic relationship between $w_{Slow}$ and the LMIT suggests that we can use the LMIT estimate as a proxy for the $w_{Slow}$ estimate, when analyzing experimental data

lagged correlations and taking the intercept of the linear fit at zero correlation (see Methods section for a more formal definition). We then transform this session lag measure to trials by multiplying it with the mean session size so that the LMIT is expressed in trials. The LMIT is a lower bound for the longest timescale of reward integrators.

We performed this analysis for the data generated by the model with different fixed relative weights of $w_{Slow}$ of the longest timescale ($\tau_{Slow} = 1000$ trials), and the results are plotted as a function of the time lag (before session-trial transformation) in Fig. 6b. The estimated LMIT is plotted in Fig. 6c as a function of the relative weight $w_{Slow}$. This plot links the model parameter $w_{Slow}$ to the model-independent measure LMIT. The LMIT is

rather sensitive to changes of $w_{Slow}$, suggesting that we can use it as a measure of the impact of slow reward integration on choice.

With this in mind, we then directly measured the LMIT in the experimental data. As predicted, we find that the LMIT covaries with the deviation from the matching law. As shown in Fig. 7a, b for both monkeys, not only does the LMIT change over time (Fig. 7a), but it also correlates with the degree of undermatching (Fig. 7b). This confirms our prediction that undermatching (bias towards a 1:1 ratio of choices) reflects very slow reward integration over hundreds or thousands of trials.

Note that the estimate of undermatching appears to be less noisy than that of the LMIT (Fig. 7a). This is due to the difference in the number of data points that could be used to estimate each

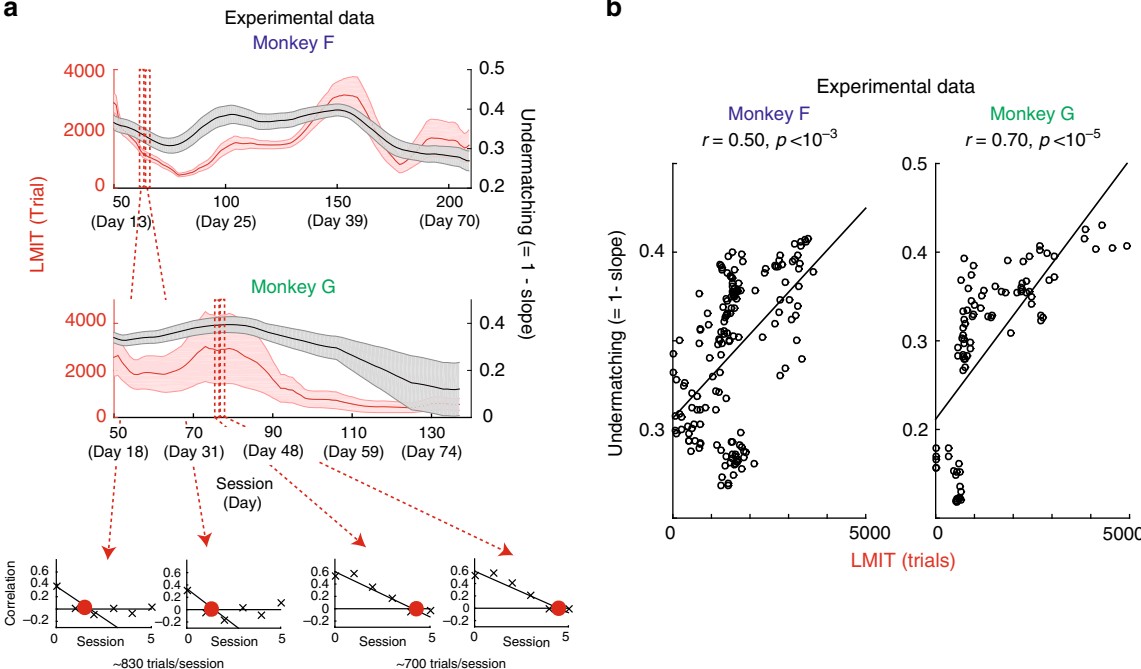

**Fig. 7** Measuring the LMIT in experimental data and showing correlations with undermatching. **a** We estimated the LMIT directly in experimental data. Both LMIT (red) and deviation from the matching law (black) vary over time for both monkeys. In the small insets at the bottom, we show four samples of the lagged correlation between color choice bias and color reward imbalance (the same plots as the ones in Fig. 6b, but now for the real data). The degree of undermatching seems to be correlated with the LMIT, as predicted by the theory. This impression is confirmed by the analysis in **b**, where we plotted the degree of undermatching as a function of the LMIT. The correlations between the two quantities are highly significant for both monkeys (conservative piece-wise permutation test: $p < 10^{-3}$ for monkey F and $p < 10^{-5}$ for monkey G). See the Methods section for the details of the permutation test. See also Supplementary Figure 14. Notice that the LMIT is a measure of integration time constant that does not require any assumption about the underlying model

variable. Undermatching is estimated by regressing the data points in a window of 25 experimental sessions (>100 data points), whereas the LMIT is estimated by regressing <10 points from cross-correlations between color reward imbalance and color choice bias.

It is important to stress that our predictions did not rely on the details of our computational model (illustrated in the previous section), such as the exact time constants for reward integrators. We conducted a model-independent analysis that is only based on the correlation between undermatching and the lower bound of the longest integration time. We note that, in general, measurements of a slope and an intercept in a linear regression can be correlated; however, we did not see a significant correlation in our analysis.

**Undermatching and harvesting performance**. Our computational analysis of the multi-timescale learning model (Fig. 2a) also predicts that deviation from the matching law should be accompanied by changes in the harvesting efficiency, as a result of the bias-variance tradeoff (see Fig. 3e). As the weight of the integrator with the longest timescale increases, the harvesting efficiency should first increase and reach a maximum and then decrease. Moreover, if $w_{Slow}$, or its proxy – the LMIT, changes over time, it should be accompanied by parallel changes in harvesting efficiency. We found that this is indeed the case, as shown in Fig. 8a, b.

**Slow integration time constant depends on experimental schedule**. The main reason to modify the relative weights of the integrators is to adapt to the volatility of the environment. In the case of the experiment, there should be no reason to modify the weights once the optimal value is found. This is because the

volatility, determined by the block size, was kept relatively constant. Nevertheless, we found that the relative weights, and analogously the LMIT, changed over the months of the experiment. Explaining these dynamics is beyond the scope of our current study as they are probably due to factors that are not under control, which might include what happens in the intervals between consecutive experimental sessions. This explanation is supported by the evidence that the LMIT changes in a way that can be related to both the interval between experimental sessions and the duration of consecutive experimental sessions (see Fig. 9). Specifically, we found that the LMIT decreased when the mean interval between experiments (recent break length) was long, suggesting a process of forgetting of the degree of volatility. Instead, the LMIT increased when recent experimental sessions were longer and contained more trials. This is also the case for the relative weight of slow learning of our model that is fitted to data (Supplementary Fig. 16).

## Discussion

Deviations from the matching law are sometimes interpreted as failures due to limitations in cognitive or perceptual systems, such as neural noise. Here we used simple computational models and the analysis of experimental data to show that these deviations may actually reflect a sophisticated strategy to deal with the variability and unpredictability of non-stationary environments. In our analysis, we linked a bias-variance tradeoff, multiple timescales of reward integrations, and undermatching behavior. The bias-variance tradeoff can manifest itself in a large class of learning models in dynamic inference tasks, including the monkey experiment that we analyzed, and the simpler task that we used to solve our model analytically. Importantly, the bias-variance tradeoff is not unique to our model with multiple time

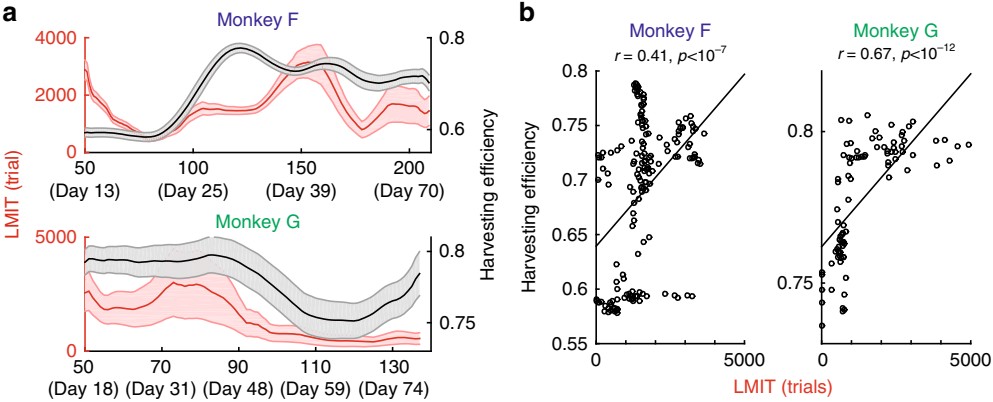

**Fig. 8** LMIT is also correlated with harvesting efficiency in data, as predicted by our modeling analysis. **a** Changes in LMIT (same as in Fig. 7) and harvesting efficiency. The harvesting efficiency was computed for each session, then taken the mean over each reference window to match the estimated LMIT. **b** The LMIT and harvesting efficiency show a significant positive correlation for both monkeys, as predicted by our theoretical analysis (The conservative piece-wise permutation test: $p < 10^{-7}$ for monkey F and $p < 10^{-12}$ for monkey G). This supports our idea that having a very slow learning of reward history, in addition to a fast one, is indeed beneficial

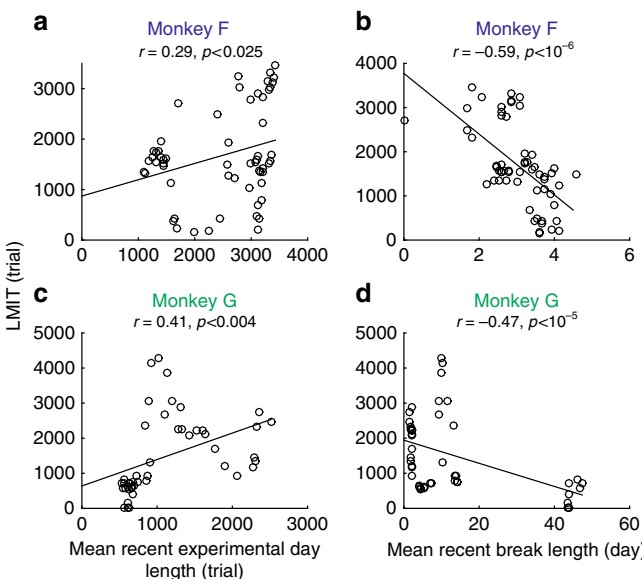

**Fig. 9** Data shows that the LMIT is affected by the length of recent consecutive experimental sessions (model-independent analysis). **a**, **c** LMIT increases as the monkeys completed more trials in recent days. The LMIT and the mean recent experimental day length (how many trials monkeys completed in one experimental day) show a significant positive correlation [permutation test: $p < 0.025$ for Monkey F and $p < 0.004$ for Monkey G]. **b**, **d** LMIT decreases as monkeys had longer breaks between experimental sessions. The LMIT and the mean recent break length show a significant negative correlation [permutation test: $p < 10^{-6}$ for Monkey F and $p < 10^{-5}$ for Monkey G]. The LMIT is computed in the same way as the previous figures with sliding windows, and both the mean recent experimental day length and the mean recent break length are computed as the average over the past 12 experimental days. Please see Supplementary Fig. 15 for the time course of changes in these variables, and also Supplementary Fig. 16 for the correlations with $w_{Slow}$, instead of LMIT

constants. In well-studied models with single learning rates, the tradeoff manifests itself as a problem in tuning learning rate (e.g. see refs. [6,7]). One can deal with the tradeoff by changing a single time constant that characterizes the learning rate. Here we used an alternative approach in which we tuned the relative weights of multiple integrators, each with a different but fixed time constant (see also the discussion below).

In fact, the bias-variance tradeoff has been extensively studied in machine learning[35]. When fitting a model to data, the ideal model should both accurately capture the observed data, and also generalize to unseen data. One way of addressing this problem is to introduce a prior belief about the data into the model, so that the model's estimate does not entirely rely on the current observation, which are corrupted by noisy processes that do not generalize to unseen data. Our present findings in monkeys could reflect a behavioral correlate of this process, as learning over a long timescale can naturally be interpreted as building a prior (see Supplementary Notes for a more mathematical discussion).

The idea of learning on multiple timescales has been suggested by previous observations on the behavior of primates[12,25] and pigeons[36], as well as other computational model studies[8,11,13]. Our work shares features of previous studies on suboptimal perceptual decisions, in which prior expectations have been shown to interfere with current perceptions[37–39]. Moreover, there has recently been accumulating evidence that several neural systems – ranging from individual neurons studied in vitro, to complex neural circuits studied in vivo – have the ability to integrate their inputs, including those that represent rewards, on multiple timescales[40–51]. Our results can provide new insights into the computational advantage of ubiquitous biological processes that operate over multiple timescales.

We showed that monkeys seem to slowly adapt relative weights of fast and slow reward integrators. We note that this meta-learning strategy is qualitatively different from, but not contradictory with, prior work on the adaptation of a single learning time-constant (e.g[6,7,9,24].), though we show in Supplementary Figure 17 that a model with a single varying timescale would be incompatible with our experimental data. Previous studies have shown that humans and animals appear to be capable of dynamically adjusting the timescale of learning. This could be achieved by changing the time constant itself, but also by adjusting the relative weights of multiple, perhaps fixed, well separated, time constants. As we discuss further below, optimizing the relative weights of multiple time-constants may be more beneficial than optimizing a single time-constant, especially when the environment changes over a wide range of timescales[8] (see also Supplementary Figure 18 and the discussion below).

For illustrative purpose we presented models with two or three time-constants; however, there may be more than three well-separated time constants that span subjects' lifetimes. In fact, previous computational studies have shown that in order to optimize memory capacity, it is beneficial to have logarithmically distributed time constants, which can efficiently span a wide range of timescales[8,14,52]. Under this scenario, the resulting reward integration kernel (the sum of exponents) becomes scale-free (power-law). In such a system, trying to determine the exact time constants of the integrators might be misleading because the scale-free integration kernel can be well-approximated by many different choices of sums of exponents with different timescales. So there may be many more timescales than what we characterized, ranging from one trial to thousands of trials, and these time constants themselves may also be adaptive. We should also note that our primary interest in this study was the role of very slow learning, over very long time constants[53]. Hence we did not analyze the effect of what is often referred to as change-point detection, where subjects detect and rapidly speed up their learning rates in response to sudden changes in reward schedules (e.g. refs. [6,7,14,54]). Although such a trial-by-trial adaptation is beyond our current scope, this can be naturally incorporated to our learning mechanism by allowing trial-by-trial adaptation of relative weights of reward integrations. Nonetheless, our analysis suggests that change-point-detection would have affected only the weights of short time-constants, as the long-term bias was present throughout the experimental sessions, which is consistent with what we have previously predicted in our neural network model analysis[8].

Integration over multiple timescales can be implemented by neural circuits in several ways, for example, by synaptic models endowed with metaplasticity[8,17,52], or by partitioned, interactive, memory systems that are responsible for preserving memories on different timescales (see e.g. ref. [15]). These models have shown to have significant computational advantages over models with a single timescale. In fact, a simple neural-network model involving synaptic metaplasticity, often referred to as the cascade model of synapses[8,52], can capture some of the key aspects of our data[55]. The model encodes the reward history over multiple timescales, as individual synapses continuously change the rate of plasticity and the effective timescales are distributed logarithmically over a broad range. As seen in Fig. 10c, the model can reproduce the changes in matching behavior observed in the experiment. In the early days of the experiment, the model shows good matching behavior, as we assume that at the beginning of day 1, the metaplastic states, which determine the plasticity rate, are distributed uniformly (e.g. ref. [8,56]). As learning progresses, the distribution of synapses in the cascade model gradually shifts as a result of memory consolidation, introducing a bias toward the mean of the distribution of rewards estimated on long timescales. Since the rewards are balanced on these timescales, the bias leads to undermatching. During long breaks between experimental sessions, however, the model forgets what it has learned, driven by activity that is uncorrelated with the reward structure of the experiment. Thus, our synaptic plasticity model can capture the observed changes in behavior through the interplay of memory consolidation (synapses become less plastic due to learning during sessions), and forgetting (synapses become more plastic during long inter-session-intervals). As shown in Fig. 10d the model also captures the bias variance tradeoff that we observed in the data.

We showed that the harvesting performance of the monkey increases when integration of reward history over much longer time scales than traditionally thought relevant is taken into account. This improvement could be significantly larger in other situations. In a two-choice task the harvesting performance varies

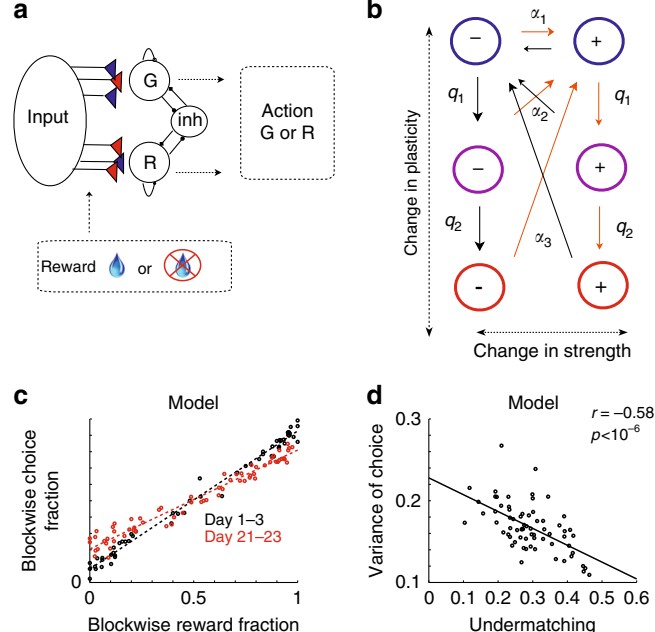

**Fig. 10** The metaplastic model of synapses (the cascade model[52]) can capture key aspects of experimental data. **a** Network model. A decision is made according to the competition between the two action selection populations (G: choosing Green target; R: choosing Red target), mediated by the inhibitory population. The competition is determined by synaptic strength between the input population and the action selection populations. These synapses are plastic, and they encode the value of Green target and Red target[8,12,20,56,61]. **b** The cascade model of synapses. Each synapse makes Markov transitions between states with a different strength (depressed −, or potentiated +) and plasticity (upper states are more plastic) with given probabilities, where transition probabilities are designed to be ordered as $\alpha_1 \gg \alpha_2 \gg \alpha_3, q_1 \gg q_2$. Note that transition probabilities are logarithmically distributed so that deeper states are harder to enter, and harder to leave. In general, the number of metaplastic states (vertical states) can be more than three[8,52]. Here we show three states for an illustrative purpose. **c** The model captures changes in matching behavior. The model was simulated in the same conditions experienced by Monkey F, according to the learning rules that allow reward-dependent transitions during experimental sessions, and transitions that incorporate forgetting during long breaks between experimental sessions[8]. **d** The model also captures the tradeoff between undermatching and the variance of choice

in a rather limited range when the behavior goes from random to optimal. This is a well-known limitation which makes it difficult to establish how close the behavior is to optimal. In more complex, but perhaps more realistic, tasks that involve multiple choices and timescales, the situation could be drastically different. Imagine, for example, a task in which there are 100 different choices and only one of them is rewarded with a probability that is significantly different from zero, using a schedule similar to the one of the two-choice task that we analyzed. The rewarding targets may change over; but within a restricted fraction of targets, say 10 targets, whose locations also change on a much slower timescale. In this case long timescales contain important information about the possible rewarding targets to consider and ignoring this information would lead to a significant decrease in the performance (see Supplementary Figure 18).

Our results also provide new insights into well-documented, suboptimal, exploratory choice behaviors in animals, and what is often referred to as the exploitation-exploration tradeoff[57,58]. It has been reported that animals often fail to exploit the optimal choice but instead show more random, matching-like, behavior in

simple slot machine or N-armed bandit tasks (e.g.[59]). Computationally, this has been often accounted for by adding noise to the value computations or decisions. Our current study, however, suggests that such apparent sub-optimal exploratory behaviors can be driven by very slow learning, or reward memory, on a very long timescale. Thus, our model predicts that it ought to be possible to manipulate such exploratory behavior by changing the reward statistics operating on long timescales, or possibly by directly stimulating the neural circuits responsible for long-timescale reward learning[46,48,60].

## Methods

**Subjects and the task**. As the full details of the experimental protocols are reported in refs. [18,25], here we only provide a brief description. We used two adult male rhesus monkeys (Macaca mulatta) weighing 7 and 12 kg. To initiate a series of trials, the animals had to fixate a black central fixation cross for a period of 300 ms. This was followed by a presentation screen, consisting of the black central fixation cross and two choice targets at a pair of mirror symmetric locations in opposite hemifields – a green circle and a red circle. The animals had to fixate a central point during this period (1–2 s). The location of the red target and the green target was counter balanced across trials. At the end of this delay period, the fixation cross became white (Go cue), signaling that the animal should indicate its choice with an eye movement to one of the two choice targets within a 1 s grace period. The animal was required to maintain its gaze on the chosen target for a further variable hold period of 300–600 ms. If the chosen target was baited at the time of the animal's choice, a fixed magnitude fruit juice reward was delivered during this hold period. At the end of the hold period the presentation screen reappeared, cueing the animal to return its gaze to the fixation cross within a grace period of 1 s in order to trigger the onset of the next trial. Trials continued as long as the animal maintained its gaze within a 2° spatial window centered on the location of the fixation cross or chosen target. A failure of maintaining a gaze lead to a 2–4 sec of a timeout period, before the animal was once again given the opportunity to fixate the central cross.

The sum of the reward-baiting probabilities on the two colored targets was set constant at ~0.35 rewards per trial. An empty target was baited with the given baiting probability. When a target was baited, a reward was delivered if the animal chose the target. This made the target empty, with no reward available for the target, until the target became baited. The relative reward-baiting probabilities on the two colors were set constant for a block of trials and changed without a signal between blocks of trials. The typical block size was about 100 trials. On each block the relative baiting probabilities on the two colors was chosen unpredictably from a subset of the ratios (though not all experiments used all ratios). Each session consisted of multiple blocks of trials, and animals performed typically several sessions per an experimental day. Experimental days were separated in a non-systematic way.

Throughout the experiments, a changeover delay (COD) was imposed. This is a common manipulation necessary to ensure matching behavior by discouraging simple alternating strategies[18]. We implemented a COD by delaying delivery of programmed rewards following switches between colors until the second consecutive choice of the new color. Monkeys learned the COD: they chose the new color twice in a row with a probability of over 95%. As second choices after switches were imposed by the COD, we excluded those choice from our analysis. We called the remaining choice as free choice. In practice, this was achieved by treating the first two choices that mark a switch between colors as a single choice.

**Defining and measuring undermatching**. There are two different choice biases that we used in our analysis: undermatching and color choice bias.

Undermatching, or deviation from matching behavior, was defined by 1 minus the slope of block-wise matching behavior. The slope of block-wise matching behavior was estimated by regressing a block-by-block fraction of choice allocated to the green target to the fraction of rewards collected from the target in a corresponding block. While in this paper we used a classical linear regression to estimate the slope, it is possible to consider Deming regression to improve the estimate of the slope.

While undermatching was defined as a bias in slope, color choice bias was defined by the intercept of matching behavior. Specifically, we defined color choice bias as the value of the fitted line at a reward fraction of 0.5. We used this bias when estimating a long timescale of reward integration (see below).

**Description of the multi-timescale learning model and its simulations**. We extended a previously introduced model[18] in which the income for each target ($I_G$ and $I_R$, for the green and red target respectively) was integrated on a single timescale. In our model we consider multiple timescales as follows. The probability

of choosing the red target is:

$$P_R = \frac{I_R}{I_R + I_G} \qquad (2)$$

In Sugrue et al.[18], the local incomes are assumed to be computed on a single timescale. Here we assume that the incomes are computed on multiple timescales in parallel:

$$I_{R,i}^t = \left(1 - \frac{1}{\tau_i}\right) I_{R,i}^{t-1} + \frac{1}{\tau_i} r^{t-1}, \qquad (3)$$

where $I_{R,i}^t$ is the local income from target R on trial $t$ ($t = 1,2,3,...$) computed over the timescale of $\tau_i$, and $r^{t-1}$ is 1 (0) when the target was rewarded (no-rewarded) at $t - 1$. The local income is a weighted sum of different timescales:

$$I_R^t = \sum_{i=1}^m w_i I_{R,i}^t, \qquad (4)$$

where the weights $w_i$'s are normalized so that $w_1 + w_2 + .. = 1$. In the model simulated in Fig. 2, the parameters were $m = 2$, $\tau_1 = \tau_{Fast} = 5$ trials, $\tau_2 = \tau_{Slow} = 10,000$ trials. In Fig. 3d, e, the model with different $w_{Slow}$ was simulated in Monkey F's reward schedule. $\tau_{Fast} = 2$, $\tau_{Slow} = 1000$ trials.

**Model-fitting**. We fit three-time-constant model presented in Fig. 5, using maximum likelihood estimates. We fit each session independently, except that we set the initial income estimates of the slow integrator as the same as the last income estimates of the slow integrator, since the slow integrator may carry over the previous estimates of the income. We, however, found that this assumption is not critical, due to the confound discussed in the manuscript. We set the time constants as $\tau_{Fast-1} = 2$ trials, $\tau_{Fast-2} = 20$ trials, and $\tau_{Slow} = 1000$ trials, though the exact choice of these do not affect our results.

**Estimating variance of choice in the data**. We computed the variance of Monkey's choice in Figs. 3d and 4 and in Supplementary Figure 12 as follows. First, monkey's choice time series was smoothed via two half-gaussian kernels with standard deviations of $\sigma = 8$ trials and $\sigma = 50$ trials, with a span of 200 trials. This gave us two time-series, a fast one with $\sigma = 8$ and a slow one with $\sigma = 50$. The faster signal represents a local estimate of probability of choice, the slower one represents the block-scale average of this local estimate. We then estimated the variance of the local estimate as the average squared distance between the fast time series and the slow time series. We showed squared root of variance in Fig. 3d for illustrative purpose.

We estimated the variance of inference in Supplementary Fig. 12 in the same fashion, using the income from the Green target in the model.

**Confounds in fitting models with very slow time constants**. We found that any slowly changing bias leads to correlations between the weights of the slow integrator and undermatching. To understand why this is the case, consider again the multi-timescale integrator model with two time constants, one fast and one much slower, as in Fig. 2. The fast integrator will generate estimates of the current values of the two alternatives, which we call $I_G^{Fast}$ and $I_R^{Fast}$ respectively. The slow integrator will generate two additional estimates: $I_G^{Slow}$ and $I_R^{Slow}$, which are almost constant in a block of trials. Then the weighted-averaged income from Green (Red) target can be expressed as

$$I_G = w_{Fast} I_G^{Fast} + w_{Slow} I_G^{Slow},$$

and

$$I_R = w_{Fast} I_R^{Fast} + w_{Slow} I_R^{Slow},$$

respectively. The probability of choosing one of the two alternatives (say G), given by the decision policy following the matching law (Eq. 1) is then:

$$P_G = \frac{w_{Fast} I_G^{Fast} + w_{Slow} I_G^{Slow}}{w_{Fast} I_G^{Fast} + w_{Fast} I_R^{Fast} + w_{Slow} I_G^{Slow} + w_{Slow} I_R^{Slow}} \qquad (5)$$

When the relative weight of the slow integrators is much smaller than that of the fast integrators, $P_G$ can be rewritten as:

$$P_G \simeq \left(1 - \frac{w_{Slow}}{w_{Fast}} \cdot \frac{I_G^{Slow} + I_R^{Slow}}{I_G^{Fast} + I_R^{Fast}}\right) \frac{I_G^{Fast}}{I_G^{Fast} + I_R^{Fast}} + \frac{w_{Slow}}{w_{Fast}} \frac{I_G^{Slow}}{I_G^{Fast} + I_R^{Fast}},$$

or equivalently

$$P_G - \frac{1}{2} \simeq \left(1 - \frac{w_{Slow}}{w_{Fast}} \cdot \frac{I_{tot}^{Slow}}{I_{tot}^{Fast}}\right)\left[\frac{I_G^{Fast}}{I_G^{Fast} + I_R^{Fast}} - \frac{1}{2}\right] + \frac{w_{Slow}}{2 w_{Fast}} \frac{\delta I^{Slow}}{I_{tot}^{Fast}} \qquad (6)$$

where we defined the total incomes $I_{tot}^{Slow}$ and $I_{tot}^{Fast}$ to be the sum of the two targets' values estimated on slow and fast timescales, respectively, and defined $\delta I^{Slow} = I_G^{Slow} - I_R^{Slow}$ to be the difference between the two targets' values estimated on the slow timescale.

Equation (6) shows that the tilt of the matching curve relating blockwise choice to blockwise reward fractions (Fig. 1b) is given by the first factor in parentheses,

and that the slope is <1. This is because the curve representing matching behavior (mostly undermatching) is obtained by fitting a line to the scatter plot of $P_G - \frac{1}{2}$ against $\frac{I_G^{\text{Fast}}}{I_G^{\text{Fast}} + I_R^{\text{Fast}}} - \frac{1}{2}$ when averaged over a block (which is a relatively fast time scale). Within an experimental session, we can neglect small changes in $I_{G/R}^{\text{Slow}}$, as well as changes in block-averaged $I_{\text{tot}}^{\text{Fast}}$.

In other words, Eq. (6) shows that the degree of undermatching is proportional to $\frac{w_{\text{Slow}}}{w_{\text{Fast}}} \cdot \frac{I_{\text{tot}}^{\text{Slow}}}{I_{\text{tot}}^{\text{Fast}}}$, which in turn is proportional to the relative weight of the slow integrator $w_{\text{Slow}}$ itself:

$$1 - \text{Slope} \propto w_{\text{Slow}}(I_G^{\text{Slow}} + I_R^{\text{Slow}}). \tag{7}$$

We thus expect a correlation between the weight of slow integration - $w_{\text{Slow}}$ - that we obtain by fitting our model to the data and the average degree of undermatching observed in an experimental session. However, this is not in itself conclusive evidence that undermatching reflects reward history integration over long timescales.

To understand why, consider an alternative model in which reward integration happens only over the fast timescale, but the estimated values of the two targets are the sum of estimates on the fast timescale, $I_{G/R}^{\text{Fast}}$ and random subjective-biases $B_G$ and $B_R$ that vary from one experimental session to the next. Note that by random, we more precisely mean biases to color-targets that are not accounted for by slow reward integration. Thus Eq. (5) is now replaced with

$$P_G = \frac{I_G^{\text{Fast}} + B_G}{I_G^{\text{Fast}} + I_R^{\text{Fast}} + B_G + B_R}, \tag{8}$$

where the relative weight terms are normalized to the bias terms. Again, as long as the biases $B_G$ and $B_R$ are small, similarly to Eq. (6), we obtain

$$P_G - \frac{1}{2} \simeq \left(1 - \frac{B_G + B_R}{I_{\text{tot}}^{\text{Fast}}}\right)\left[\frac{I_G^{\text{Fast}}}{I_G^{\text{Fast}} + I_R^{\text{Fast}}} - \frac{1}{2}\right] + \frac{B_G - B_R}{2I_{\text{tot}}^{\text{Fast}}} \tag{9}$$

and in particular

$$1 - \text{Slope} \propto B_G + B_R. \tag{10}$$

Fits of this alternative model would also produce correlations between $B_G + B_R$ and the average undermatching observed in experimental sessions.

Since the slow estimates $I_{G/R}^{\text{Slow}}$ in Eq. (5) are approximately constant over each experimental session (as $\tau_{\text{Slow}}$ is much longer than the experimental block length), it is difficult to discriminate the slow estimates $I_{G/R}^{\text{Slow}}$ in Eq. (5) from the random bias $B_{G/R}$ in Eq. (9). This means that we cannot discriminate these two models from model-fitting to each experimental session. In other words, although our model-independent analysis confirmed our prediction of bias-variance tradeoff in the data (Fig. 4), direct model fitting to separate experimental sessions is not sufficient to prove our other central prediction, that of a link between slow reward integration and undermatching.

**The longest measurable integration timescale**. The aim is to determine whether past biases in rewards could affect the present choice of the animal. We computed the lagged cross correlation between color reward imbalance and color choice bias. The reward color imbalance over $N$ trials is defined as $\frac{R_G - R_R}{N}$, where $R_G$ ($R_R$) is the total number of rewards that were collected from Green (Red) and $N$ is the total number of trials. The color choice bias is the intercept $y$ with $x = 0.5$ of the linear fit of matching behavior. Then we took a sliding reference window of 25 sessions. For each sliding window, we computed the correlation between color choice bias computed over 25 sessions and the color reward imbalance computed over a 25 sessions lagged in time (lag = 0, −1, −2, −3, −4, and −5 sessions). These correlations were fitted by weighted least squares $S = \sum w_i r_i^2$ with weights $w_i = \gamma^i$, where $i = 0,1,2..$ was the lag, as the correlations were more reliable for smaller lags. We used $\gamma = 0.5$ but we found that our results were robust against changes in $\gamma$. We defined the point in which the fitted line crosses the time lag axis (or 0 correlation) as the raw maximum correlation lag. Then the longest measurable integration timescale (LMIT) was then expressed as this lag multiplied by the mean session length (in trials) of 25 sessions in the reference window. The significance of the correlation between under-matching and the LMIT was determined by a conservative piece-wise permutation test which supposedly destroys the original correlations. More specifically, we considered blocks of five consecutive sessions and shuffled the order of blocks without perturbing the order of sessions within each block. This allowed us to create shuffled data with the original long timescale correlations. This was because observed LMITs were normally within five sessions. Figure 7c is smoothed by computing a moving average of 10 windows but Fig. 7d is raw data. First 25 sessions were excluded in order to discard potential effects due to training.

**Simulations of a neural-network model with metaplastic synapses**. We considered a well-studied neural network model for decision-making (Fig. 10a)[8,12,20,56,61]. Essentially, this network produces bi-stable attractor dynamics (winner-take-all process), where each stable state corresponds to the choice of Green (when G population wins) or the choice of Red (when R population wins). Crucially, the competition is

determined by the synaptic weights between the input population and the decision populations. The weights are trained by a reward based stochastic Hebbian learning that has been shown to nicely capture reinforcement-learning behavior[8,12,20,56,62].

Each of the synaptic weights is assumed to take one of the two strengths (weak or strong), as this obeys the biophysical constraint of bounded strength. In addition to the changes in efficacy, here we allowed metaplastic transitions so that synapses can change the rate of plasticity itself [8,52] (Fig. 10b). This cascade model of synapses can incorporate a various chemical cascade processes taking place over multiple timescales. It has been shown that the model improves the memory performance of bounded synapse models, and also that the model reproduces a well-observed power-low memory decay in time. Also, the model can capture a wide range of behavioral properties that has been observed in reward learning tasks, including adaptive learning rates (e.g. refs. [6,7]), and it can perform as good as a Bayes optimal model if coupled with a surprise detection network (please see[8]). We followed the previously introduced implementation of this model in decision-making tasks[8]. In particular, we assumed a forgetting between experimental sessions, implemented by random transitions between weak and strong states between sessions. We simulated our model under the same conditions as Monkey F, and we applied the same analysis to the simulated data as we did to the real data. We note that we do not implement the surprise detection system in our simulation during sessions that has been previously introduced to capture increases in learning rates at block changes[8], as our focus here was to capture behavioral changes on a longer timescale.

## Data availability

The data and code that support the findings of this study are available from the corresponding author upon reasonable request.

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

## Acknowledgements

We thank P. Dayan, L.F. Abbott, K.D. Miller, C. R. Gallistel, and K. Lloyd for fruitful discussions. Supported by NSF's NeuroNex program award DBI-1707398, the Gatsby Charitable Foundation, the Simons Foundation, the Schwartz foundation, the Kavli foundation, the Japan Society for the Promotion of Science, the Israel Science Foundation (Grant No. 757/16), the National Eye Institute, and the Howard Hughes Medical Institute.

## Author contributions

K.I., Y.A., Y.L., W.T.N., and S.F. conceived the current project. L.P.S., G.S.C., and W.T.N. have designed and run the original macaque experiment. K.I., Y.A., Y.L. and S.F. developed the theoretical models and analyzed the data, with inputs from L.P.S., G.S.C. and W.T.N.; All authors participated in writing the manuscript.
