## [Peer Review File · Nature Communications]

Reviewers' Comments:

Reviewer #1:

Remarks to the Author:

Matching law is widely seen in foraging behavior of various animal species and usually is the optimal behavioral strategy. However, the actual choice behavior of animals often deviates from matching in a non-stationary environment (undermatching). The authors proposed that such deviations reflect the fact that the brain of animals integrates reward history on multiple timescales, typically one fast and one slow timescale, and makes a decision according to a choice probability calculated from the weighted sum of the two integrations. They constructed an empirical mathematical model, in which the relative contributions of the two integrators changes with changes in the bating rates of options to optimize the efficiency of harvesting behavior: The fast integrator calculates an optimal choice probability (i.e., matching) in the current trial block while the slow integrator contributes a constant bias factor to the choice probability. The model accounts for undermatching behavior and predicts a trade-off between the bias (controlled by the slow integrator) and variance (controlled by the fast integrator) in choice behavior. Behavioral data from two monkeys were used to validate the model's predictions.

The topics of the study are important and the results are interesting. However, I have some concerns about the heuristic model proposed in this study, the methodology for its validation and the ways the results are presented.

Major comments:

1. It took me quite some time to understand why the slow timescale is needed to optimize the efficiency of harvesting behavior. If an integrator with a single fast timescale could always evaluate the current bating rates with sufficiently small variances, such a system would produce matching (optimal) behavior with small behavioral variance and hence would be an optimal integrator. In Supplementary S1, however, the authors analytically showed that this does not occur due to inherent noise in the task, and summarized the analysis results in Fig. 3. Then, in Fig. 7 the authors analyzed experimental data of monkeys to show the behavioral variances predicted by the variance-bias trade-off. I feel that Figure 7 should be shown earlier in the manuscript as the results provide a direct support for the presence of two (or more) timescales. I think that these results are the major message of this paper, when evidence for the specific model proposed remains weak (comment 4). Because the behavioral variance depends on the ratio of bating rates in each trial block, I wonder whether the authors can estimate the fast and slow time constants, or at least the ratio between them, from the behavioral data. Such analyses will strengthen the manuscript.

2. The use of the longest measurable timescale (LMIT) for experimental validations and the related explanations on page 12 are confusing. The authors introduced LMIT as a "model-independent" measure of slow integration times. However, in Figure 5 LMIT was estimated for the proposed model by simulating its behavior on the same reward schedule as the monkeys. If I understood correctly, the authors could have estimated LMIT directly from monkey data without using the model and investigated correlations between undermatching and LMIT. I wonder why the authors did not do that.

3. Indeed, the LMIT estimated by the authors' model and the deviation from matching behavior evolve quite differently in Fig. 5c: While the former shows a peaky behavior, the latter changes only gradually. Is this discrepancy caused simply by an inadequate choice of the weight or the time constant of slow integrator? Related to this point, how was the value of slow time constant determined in this study?

4. Previous studies suggested an alternative mechanism in which the adaptation of a single integration time constant occurs across trials. Though the single-time-constant model cannot explain undermatching, we can consider another model having multiple modifiable time constants that cover different ranges of timescale. The authors are aware of this point and mentioned that experimental validations of the explicit adaptation mechanisms are difficult as the slow timescale is too slow. However, it would significantly strengthen the paper, even without experimental validations, if the authors could argue for the advantage of their model over the other models.

Minor comments:

5. The words "reward color bias" and "reward bias" or "choice color bias" and "choice bias" are used for the same meanings. This confused me a bit. I feel that "color" does not play a specific role here and may be removed.

6. The authors suggested their cascade synapse model as a possible adaptation mechanism (Figure S7). The authors are perhaps interested in emphasizing this possibility in the main text.

7. At lines 163-164: ... the local incomes for the two targets (Methods).  Please refer to the Methods section.

Reviewer #2:

Remarks to the Author:

In this study, the authors analyze behavioral data from two monkeys performing a choice task. In the task, two rewards are baited at differing rates that change across blocks. Under assumptions of stationarity, the optimal strategy is matching behavior. The authors argue that observed undermatching behavior is the result of integration of rewards over long time scales, a feature that reduces variance but increases bias. They show that their model is sufficient to explain undermatching and attempt to show that a model-agnostic measure confirms some key assumptions.

I am in general favorably disposed toward the paper. It's a nice contribution to the literature on behavioral consequences of nonstationarity (or the persistent assumption of it) on behavior. Nonetheless, I am confused by some of the logic of the authors' argument, and would appreciate clarification before I could endorse publication.

I felt that I understood the argument well up to around line 258. For the rest of the section, this is my understanding of the argument as the authors are constructing it:

1. It is impossible to distinguish between a weighted inclusion of a slow integrator (w_{slow}) and a bias in the value of each target (b) on intra-session timescales. That is, the authors' model is sufficient to explain undermatching, but not necessary (?).
2. There is a correlation between choice bias to each target and the differential between the number of rewards obtained from each target over multi-session timescales (LMIT). This effect is independent of undermatching.
3. In other words, at least one aspect of monkeys' behavior (the LMIT) shows correlations over very long times, though this aspect is not the focus of the study.
4. If the model is true, this is sufficient to explain the LMIT effect.

A couple of notes before getting to the substance of this: First, the definition of "reward color bias" only occurs in methods, and the term is used without explanation in the main text. This was confusing to me. Second, as best I understand it, the LMIT correlation measures the ability of a mismatch in

number of rewards obtained from each color target to influence bias toward each color many trials later. So just to be clear: it's color bias that lags? And the assumption is that reward color bias is averaged over a long enough time period that the colors are approximately equal in value (that is, enough switches in reward rate have taken place)? I believe this is what the authors are attempting to (compactly) explain in ll. 263-265, but that wasn't sufficient for me to follow the argument the first time through.

Now, my main confusion is what, exactly, the authors are trying to establish. Is it simply that monkeys' behavior can, in fact, show correlations over such long time scales, arguing that slow integration is plausible? Or is it that the LMIT analysis is somehow proving the correctness of the authors' model? The integrator model seems reasonable to me, so I'm just missing something about what question the LMIT analysis is supposed to answer.

Related to this: I'm perhaps also a bit unclear about what precise point is being made on lines 243-257. I interpreted this (and the following equations) as a demonstration that for very slow integration, the correlation in 4c is equivalent to a bias in the integrator estimates, and so does not provide direct evidence for integration.

But along the same lines, if I consider the next order approximation:

$$I^{\text{Slow}}_G = b + \delta$$

$$I^{\text{Slow}}_R = b - \delta$$

with $\delta \ll b$ (that is, $b = (I^{\text{Slow}}_G + I^{\text{Slow}}_R)/2$), then the expansion for P_G to next order is

$$P_G \approx$$

$$\left(1 - \frac{2b}{I^{\text{Fast}}_G + I^{\text{Fast}}_R}\right) \frac{I^{\text{Fast}}_G}{I^{\text{Fast}}_G + I^{\text{Fast}}_R} + \frac{b + \delta}{I^{\text{Fast}}_G + I^{\text{Fast}}_R}$$

$$P_G \approx$$

In this case, doesn't the δ term on the right-hand side work out to a color choice bias?:

$$P_G \approx$$

$$\left(1 - \frac{2b}{I^{\text{Fast}}_G + I^{\text{Fast}}_R}\right) \left(\frac{I^{\text{Fast}}_G}{I^{\text{Fast}}_G + I^{\text{Fast}}_R} - \frac{1}{2} - \frac{\delta}{I^{\text{Fast}}_G + I^{\text{Fast}}_R}\right) + \frac{1}{2} - \frac{\delta}{I^{\text{Fast}}_G + I^{\text{Fast}}_R}$$

$$P_G \approx$$

That is, if we include a previous reward color bias in b , can't we explain the LMIT effect as well?

Again, I seem to be missing a piece of the logic here.

Small notes:

l. 240: fitted fit

l. 282: should be "not only does the LMIT change over time"

Reviewer #3:

Remarks to the Author:

In this article, authors investigate learning choices in a probabilistic bandit task in monkeys. Under the assumption that the contingencies are stable, a matching behavior would be optimal. However, here, the contingencies change abruptly, and the monkeys exhibit undermatching. The authors use computational modeling to show that this behavior leads to more rewards for the animals, and that it reflects two time scales of integration of information – a slow time scale reflected by bias in choices, and a fast time scale visible in undermatching (weaker dependence on reward than expected).

This is an interesting and well written paper, and I only have a few concerns. My first concern is in the framing of the paper in terms of optimality, in particular in the introduction. My second concern regards the model-based analysis of the behavioral data.

The introduction is framed in terms of optimal behavior, which matching is for the stable version of the task used here. It makes no sense to insist on this so much in the introduction, and to present undermatching as "surprisingly unoptimal" here, since the task is different: reward probabilities change over time unpredictably. Statements like "Paradoxically" (l131), and "This observation seems to be incompatible with the statement that matching 133 behavior is optimal in this task" (l133) are artificial and unhelpful. By definition, the task is different with changes over time, and so it is unsurprising that the optimal behavior is different. Furthermore, this highlights how important assumptions about the knowledge of the task environment are when attempting to define "optimal" behavior, and thus how hard to interpret any claims of optimality (or suboptimality) in behavior are. In my opinion, this paper would be much better focused on the existence of multiple time scales of integration in learning, rather than on whether this is "optimal" or not.

My second main concern is about the model-based analysis of the data. I have two sub concerns – first, that a number of parameters are "hand-chosen", second, that the model fit is done on aggregated rather than raw data (to the best of my understanding of the methods).

The authors smooth the choices with arbitrarily chosen kernels (l646-648), then fit a curve to the obtained modified data. Why do the authors not do a trial-by-trial fit of the model with the data, optimizing the log likelihood of choices across the range of relevant parameters, instead? The current procedure likely loses some of the local structure of the data by smoothing, for no clear reason. There are perfectly valid tools for trial by trial analysis of models, and the proposed model is amenable to them (e.g. see multiple didactic papers by Daw and colleagues, Palminteri et al 2017, etc). These methods would have the advantage of 1) benefitting from all the behavioral richness of choices 2) fitting, rather than choosing key parameters 3) allowing simpler model comparison.

Minor concern:

There are some minor language issues in the Methods section:

- l619 raw -> row
- l621, l633: "we call XX as"

Reviewer #1

Matching law is widely seen in foraging behavior of various animal species and usually is the optimal behavioral strategy. However, the actual choice behavior of animals often deviates from matching in a non-stationary environment (undermatching). The authors proposed that such deviations reflect the fact that the brain of animals integrates reward history on multiple timescales, typically one fast and one slow timescale, and makes a decision according to a choice probability calculated from the weighted sum of the two integrations. They constructed an empirical mathematical model, in which the relative contributions of the two integrators changes with changes in the bating rates of options to optimize the efficiency of harvesting behavior: The fast integrator calculates an optimal choice probability (i.e., matching) in the current trial block while the slow integrator contributes a constant bias factor to the choice probability. The model accounts for undermatching behavior and predicts a trade-off between the bias (controlled by the slow integrator) and variance (controlled by the fast integrator) in choice behavior. Behavioral data from two monkeys were used to validate the model's predictions.

The topics of the study are important and the results are interesting. However, I have some concerns about the heuristic model proposed in this study, the methodology for its validation and the ways the results are presented.

Major comments:

1. It took me quite some time to understand why the slow timescale is needed to optimize the efficiency of harvesting behavior. If an integrator with a single fast timescale could always evaluate the current bating rates with sufficiently small variances, such a system would produce matching (optimal) behavior with small behavioral variance and hence would be an optimal integrator. In Supplementary S1, however, the authors analytically showed that this does not occur due to inherent noise in the task, and summarized the analysis results in Fig. 3. Then, in Fig. 7 the authors analyzed experimental data of monkeys to show the behavioral variances predicted by the variance-bias trade-off. I feel that Figure 7 should be shown earlier in the manuscript as the results provide a direct support for the presence of two (or more) timescales. I think that these results are the major message of this paper, when evidence for the specific model proposed remains weak (comment 4).

Our reply:

Thanks to the comments by the reviewers, we realized that the previous version of the manuscript lacked clarity in many important parts, and in the overall logic of our analysis. Showing Figure 7 earlier can certainly help clarify the logic of our manuscript; we appreciate the suggestion. We have moved (old) Figure 7 to (new) Figure 4, and revised the text accordingly.

Because the behavioral variance depends on the ratio of bating rates in each trial block, I wonder whether the authors can estimate the fast and slow time constants, or at least the ratio between them, from the behavioral data. Such analyses will strengthen the manuscript.

Our reply:

A previous analysis of the same experimental data focusing on monkey's fast time constants (Corrado et al., 2005) has suggested that we need at least two time constants to characterize monkey's fast reward integration. We confirmed this by fitting

multiple exponents to the auto-correlation of monkey's choice history (please see Figure S1a). One of the monkeys (Monkey F) showed time constants of approximately 1 and 43 trials, while the other monkey (Monkey G) showed time constants of approximately 1 and 42 trials. Our LMIT analysis then showed that the monkeys also had much slower time constants for slow reward integration, which are longer than 1,000-4,000 trials (please see new Figure 7ad). From these analysis, we could estimate the ratio between the short, and the long, time constants as large as 1:100 or 1:1000.

Notice, however, that the LMIT analysis provides us with a lower bound on the longest measurable time constant. Thus there may be time constants that are even longer than our LMIT estimates, or there may also be intermediate times constants that could not be parsed by our LMIT analysis. These are because our LMIT analysis only estimates "effective" time constants, as we showed using simulated data that LMIT estimates would vary according to changes in relative weights of slow and fast integrators of our model generating the data (please see new Figure 6bc). We should also note that fitting a learning model with long time constants to each experimental session does not provide reliable estimates of time constants for reward integration, since each session is too short for estimating slow reward integration and its effects on choice, as we now have discussed in more detail in our manuscript.

In fact, previous studies have suggested computational advantages of reward integration over multiple time constants, which are significantly more than three (e.g. Fusi et al. 2005). These studies suggest the resulting reward integration kernel would become scale free (e.g. a power law, such as $1/(1+t)$, where t is the number of trials), since the sum of a large number of multiple exponential integrators would become approximately scale free (e.g. Fusi, et al. 2005). If this is the case, estimating precisely each time constant would be fruitless, because time constants are not uniquely determined.

Nonetheless, we should note that our primary focus was on computational advantages, and behavioral consequences, of slow reward learning, and that our main results do not rely on precise estimates of short and long time constants. Thus we did not aim to validate a specific model (such as one of Figure 2a); but instead, we confirmed the evidence for (so-far largely-ignored) computational benefit of reward integration on very long time constants.

To clarify all these points in the manuscript, we expanded the Discussion:

(line 443) For illustrative purpose we presented models with two or three time-constants; however, there may be more than three well-separated time constants that span subjects' lifetimes. In fact, previous computational studies have shown that in order to optimize memory capacity it is beneficial to have logarithmically distributed time constants, which can efficiently span a wide range of timescales[17, 64, 22]. Under this scenario, the resulting reward integration kernel (the sum of exponents) becomes scale-free (power-law). In such a system, trying to determine the exact time constants of the integrators might be misleading because the scale free integration kernel can be well-approximated by many different choices of sums of exponents with different timescales. So there may be many more timescales than what we characterized, ranging from one trial to thousands of trials, and these time constants themselves may also be adaptive. We however note that what we showed by our model-independent analysis is consistent with this more general models with many, adaptive,

time-constants, as the LMIT analysis does not discriminate if the relative weights, or the time constants, are adaptive.

(line 418) Our goal was not to validate a specific computational model, such as the one of Figure 2a, but instead to confirm the existence and benefit of, reward integration over very long timescales in a dynamic, reward-based, learning task.

We now have expanded the explanation for why fitting a reward integration model does not provide sufficient evidence for slow time constants and their effects on choice (i.e. undermatching behavior).

(line 285) Further, as we predicted, changes in the weight of the slow integrator are correlated with changes in undermatching (Figure 5). However, due to a potential confound, this correlation is not sufficient to prove the link between undermatching and slow reward integration. This is because the slow reward integrators act as a bias that changes slowly over time, and (as we will show below) any slowly changing bias will lead to correlations with undermatching, regardless of whether that bias is driven by slow reward integration or some other process unrelated to reward history. For example, a bias that is randomly modified from session to session would also show correlations with undermatching, even though that bias does not depend on reward history.

Please also see the paragraph following the above in the results section for the detailed, mathematical, argument.

2. The use of the longest measurable timescale (LMIT) for experimental validations and the related explanations on page 12 are confusing. The authors introduced LMIT as a "model-independent" measure of slow integration times. However, in Figure 5 LMIT was estimated for the proposed model by simulating its behavior on the same reward schedule as the monkeys. If I understood correctly, the authors could have estimated LMIT directly from monkey data without using the model and investigated correlations between undermatching and LMIT. I wonder why the authors did not do that.

Our reply:

We apologize for the confusion. As a matter of fact, we estimated the LMIT directly from the monkey data, without referring to (or fitting) specific generative models. Panels a and b of Previous Figure 5 (now Figure 6,c and d) were supposed to simply establish a link between the weight of the long time constant in the simplified model (that we used to explain the bias-variance tradeoff) and the LMIT. The LMIT is an independent quantity that should be measured directly from data, thus we first showed a link between this model-independent measure to our computational model, using simulated data from the computational model. Then we estimated the LMIT directly from the simulated data (previous Figures 5 cd; now new Figure 7ab), without using any model. We now have clarified this in our manuscript and in the new Figure 6a:

(line 373) With this in mind, we then directly measured the LMIT in the experimental data. As predicted, we find that the LMIT co-varies with the deviation from the matching law. As shown in Figure 7b for both monkeys, not only does the LMIT change over time (Figure 7), but it is also correlated with the degree of undermatching (Figure 7). This confirms our prediction that undermatching (bias towards a 1:1 ratio of choices) reflects very slow reward integration over hundreds of, or thousands of, trials.

3. Indeed, the LMIT estimated by the authors' model and the deviation from matching behavior evolve quite differently in Fig. 5c: While the former shows a peaky behavior, the latter changes only gradually. Is this discrepancy caused simply by an inadequate choice of the weight or the time constant of slow integrator? Related to this point, how was the value of slow time constant determined in this study?

Our reply:

In our previous version, both the LMIT and deviations from matching behavior in previous figure 5c (now new Fig 7a) were estimated directly, independently, from the experimental data. Sorry again for the confusion. We clarified this in the text. As seen in original figure 5d (now new figure 6d), the LMIT and undermatching showed strong correlations in both monkeys, which we predicted from our theoretical analysis. Undermatching may look varying more smoothly than the LMIT; this may be due to the difference in the number of data points that are used to estimate each variable. Undermatching is estimated by a regression of data points in each block of trials with fixed reward contingencies (>100 points in each 25 session window), while the LMIT is estimated from limited points from cross-correlations over sessions (< 10 points in each 25 session window). Thus the LMIT is expected to be noisier than undermatching. We added to the text the following sentences that comments on these difference in the speed of variation of the two quantities:

(line 379) Note that the estimate of undermatching appears to be less noisy than that of the LMIT (Figure 7). This is due to the difference in the number of data points that could be used to estimate each variable. Undermatching is estimated by regressing the data points in a window of 25 experimental sessions (> 100 data points), whereas the LMIT is estimated by regressing less than 10 points from cross-correlations between color reward imbalance and color choice bias.

4. Previous studies suggested an alternative mechanism in which the adaptation of a single integration time constant occurs across trials. Though the single-time-constant model cannot explain undermatching, we can consider another model having multiple modifiable time constants that cover different ranges of timescale. The authors are aware of this point and mentioned that experimental validations of the explicit adaptation mechanisms are difficult as the slow timescale is too slow. However, it would significantly strengthen the paper, even without experimental validations, if the authors could argue for the advantage of their model over the other models.

Our reply:

We agree that multiple time constants themselves can also be adaptive. From a computational view point, however, such a model with variable time constants, and models with fixed time constants with variable weights, are in many ways equivalent. As we mentioned in our response to the reviewer's first comment, the difference is in fact artificial, because a model with variable time constants can be mapped onto a model with fixed time constants with variable weights. This is most apparent in the case of scale-free reward kernel. Thus it was beyond our scope of the manuscript to tease apart details of slow adaptation mechanism.

Our primary interest was instead to show the existence of a very long time constant that has been previously largely ignored, and to explain its computational role in decision-making. For this purpose, both of these sub-classes of models (models with fixed time constants with variable weights, and models with variable time constants) can serve as good examples for what we are interested in, as both models would be benefited from slow reward integration, and predict the same behavioral consequences in our task.

We expanded the discussion of this point in the Discussion:

(Line 443) For illustrative purpose we presented models with two or three time-constants; however, there may be more than three well-separated time constants that span subjects' lifetimes. In fact, previous computational studies have shown that in order to optimize memory capacity it is beneficial to have logarithmically distributed time constants, which can efficiently span a wide range of timescales [17, 64, 22]. Under this scenario, the resulting reward integration kernel (the sum of exponents) becomes scale-free (power-law). In such a system, trying to determine the exact time constants of the integrators might be misleading because the scale free integration kernel can be well-approximated by many different choices of sums of exponents with different timescales. So there may be many more timescales than what we characterized, ranging from one trial to thousands of trials, and these time constants themselves may also be adaptive. We however note that what we showed by our model-independent analysis is consistent with this more general models with many, adaptive, time-constants, as the LMIT analysis does not discriminate if the relative weights, or the time constants, are adaptive.

We should also note that our primary interest in this study was the role of very slow learning, or very long time constants [13]. Hence we did not analyze the effect of what is often referred to as 'change-point detection', where subjects detect and rapidly speed up their learning rates in response to sudden changes in reward schedules (e.g. [2, 6, 37, 64]). Although such a trial-by-trial adaptation is beyond our current scope, this can be naturally incorporated to our learning mechanism by allowing trial-by-trial adaptation of relative weights of reward integrations. Nonetheless, our analysis suggests that change-point-detection would have affected only the weights of short time-constants, as the long-term bias was present throughout the experimental sessions, which is consistent with what we have previously predicted in our neural network model analysis [22].

(Line 504) In fact, if one fits a reward integrator model characterized by a single timescale to the data that we analyzed, the estimated timescale that one observes is approximately optimal [55]. Additional analysis, however, has suggested that at least two, relatively short, timescales (approximately 2 and 20 trials) are required to characterize the reward integrators [12]. Importantly, the mechanism that we studied operates on much longer timescales and probably complements those that have been analyzed in the past. Our analysis cannot determine whether the LMIT changes because the relative weights of different integrators are modified [12, 17, 22, 7], or because the intrinsic time constants of the integrators change. This is because the timescale that we consider is extremely long. It is very possible that both types of models of the LMIT can accurately describe the data and capture the correlations between the LMIT and undermatching. Moreover, it is difficult to study the interaction between the mechanisms that we studied and those that determine the optimal short integration timescales. It is possible that the long timescales make the effective timescales of the fast integrators longer, but it is also possible that the mechanisms implementing the two

types of integrators are completely independent and that they both contribute to the final harvesting efficiency.

Minor comments:

5. The words "reward color bias" and "reward bias" or "choice color bias" and "choice bias" are used for the same meanings. This confused me a bit. I feel that "color" does not play a specific role here and may be removed.

Our reply:

Thank you for this comment, and we're sorry for not being clear about definitions of very similar terms. There is no difference between "reward color bias" and "reward bias". Now we refer to them as "color reward imbalance", which we define as the imbalance (difference) between the number of rewards collected for two color targets, divided by the number of trials.

(Figure 6 caption) We also estimate session-by-session color reward imbalance that we defined as the imbalance between rewards obtained from two color targets.

As for the choice bias, we now use "color choice bias" as the intercept of matching behavior, which contrasts with undermatching, defined by the slope.

(Line 153) Note that there is another well-documented deviation from the matching law [4], which we refer to as color choice bias. This is a bias toward one of the colored targets, and can be quantified as the deviation of the best linear fit to the behavior data from the unity diagonal predicted by the matching law at a reward fraction of 0.5 (see the inner panel of Figure 1 and Methods section). It is important to distinguish two biases: undermatching (slope) and color choice bias (intercept), as these are independent behavioral measures.

6. The authors suggested their cascade synapse model as a possible adaptation mechanism (Figure S7). The authors are perhaps interested in emphasizing this possibility in the main text.

Our reply:

We appreciate this comment. In fact, our project was inspired by our computational analysis of the cascade model in this task. Following the reviewer's suggestion, we now have moved (old) Figure S7 to new Figure 10, and added discussions and descriptions to the manuscript.

(line 470) In fact, in simulation studies we found that a simple neural-network model involving synaptic metaplasticity, often referred to as the cascade model of synapses [17, 22], can capture some of the key aspects of our data [25]. The model encodes the reward history over multiple timescales, as individual synapses continuously change the rate of plasticity and the effective timescales are distributed logarithmically over a broad range. As seen in Figure 10, the model can reproduce the changes in matching behavior observed in the experiment. In the early days of experiment, the model shows good matching behavior, as we assume that at the beginning of day 1, the metaplastic states, which determine the plasticity rate, are distributed uniformly (e.g. [54, 22]). As learning progresses, the distribution of synapses in the cascade model gradually shifts as result of memory consolidation, introducing a bias

toward the mean of the distribution of rewards estimated on a long timescale. Since the rewards are balanced on the long timescale, the process is manifest as seen as deviation from the matching law, or undermatching. During long breaks between experimental sessions, however, the model forgets what it has learned, driven by activity that is uncorrelated with the reward structure of the experiment. Thus, our synaptic plasticity model can capture the observed changes in behavior through the interplay of memory consolidation (synapses become less plastic due to learning during sessions), and forgetting (synapses become more plastic during long inter-session-intervals). As shown in Figure 10 the model also captures the bias variance tradeoff that we observed in the data.

Also in the Methods section:

(line 787) We considered a well-studied neural network model for decision-making (figure 10) [60, 54, 16, 24, 22]. Essentially, this network produces bi-stable attractor dynamics (winner-take-all process), where each stable state corresponds to the choice of Green (when G population wins) or the choice of Red (when R population wins). Crucially, the competition is determined by the synaptic weights between the input population and the decision populations. The weights are trained by a reward based stochastic Hebbian learning that has been shown to nicely capture reinforcement-learning behavior [52, 54, 16, 24, 22].

Each of the synaptic weights is assumed to take one of the two strengths (weak or strong), as this obeys the biophysical constraint of bounded strength. In addition to the changes in efficacy, here we allowed metaplastic transitions so that synapses can change the rate of plasticity itself [17, 22] (Figure 10). This cascade model of synapses can incorporate a various chemical cascade processes taking place over multiple timescales. It has been shown that the model improves the memory performance of bounded synapse models, and also that the model reproduces a well-observed power-law memory decay in time. Also, the model can capture a wide range of behavioral properties that has been observed in reward learning tasks, including adaptive learning rates (e.g. [6, 37]), and it can perform as good as a Bayes optimal model if coupled with a surprise detection network (please see [22]). We followed the previously introduced implementation of this model in decision-making tasks [22]. In particular, we assumed a ‘forgetting’ between experimental sessions, implemented by random transitions between weak and strong states between sessions. We simulated our model under the same conditions as Monkey F, and we applied the same analysis to the simulated data as we did to the real data. We note that we do not implement the surprise detection system in our simulation during sessions that has been previously introduced to capture increases in learning rates at block changes [22], as our focus here was to capture behavioral changes on a longer timescale.

7. At lines 163-164: ... the local incomes for the two targets (Methods). – Please refer to the Methods section.

Our reply:

Thanks for this suggestion. We added the reference to the Methods section.

Reviewer #2

In this study, the authors analyze behavioral data from two monkeys performing a choice task. In the task, two rewards are baited at differing rates that change across blocks. Under assumptions of stationarity, the optimal strategy is matching behavior. The authors argue that observed undermatching behavior is the result of integration of rewards over long time scales, a feature that reduces variance but increases bias. They show that their model is sufficient to explain undermatching and attempt to show that a model-agnostic measure confirms some key assumptions.

I am in general favorably disposed toward the paper. Its a nice contribution to the literature on behavioral consequences of nonstationarity (or the persistent assumption of it) on behavior. Nonetheless, I am confused by some of the logic of the authors argument, and would appreciate clarification before I could endorse publication.

Our reply:

Thank you for the positive opinion about our manuscript, and we apologize again for the lack of clarity. We hope that the new version of our manuscript is clearer.

I felt that I understood the argument well up to around line 258. For the rest of the section, this is my understanding of the argument as the authors are constructing it: 1. It is impossible to distinguish between a weighted inclusion of a slow integrator (*w_{slow}*) and a bias in the value of each target (b) on intra-session timescales. That is, the authors model is sufficient to explain undermatching, but not necessary (?).

2. There is a correlation between choice bias to each target and the differential between the number of rewards obtained from each target over multi-session timescales (LMIT). This effect is independent of undermatching. 3. In other words, at least one aspect of monkeys behavior (the LMIT) shows correlations over very long times, though this aspect is not the focus of the study. 4. If the model is true, this is sufficient to explain the LMIT effect.

Our reply:

Our computational model analysis (new Figures 2,3) predicted triad relationships between the bias-variance tradeoff, slow reward integration, and undermatching behavior. The goal was to confirm this in experimental data.

First we validated a link between bias-variance tradeoff and undermatching in the experimental data, without resorting to a specific model-fitting (new Figure 4).

We then aimed to validate a link between these tradeoff/undermatching and slow reward integration in the data; but we found that direct model-fitting cannot sufficiently address this issue, partly due to the limitation that experimental sessions were too short to confirm such slow reward integration and its effects on choice.

So we instead introduced a model-independent analysis, using the longest measurable integration timescale (LMIT). The logic is now illustrated in new Figure 6a. First we established a link between the LMIT and the reward integrator model (new Figure 6bc), by estimating the LMIT in hypothetical data simulated from the model. Then we estimated the LMIT directly in the experimental data, without model-fitting (because the LMIT is a model-independent measure). Then we showed that the LMIT estimated

from the experimental data is indeed correlated with the degree of undermatching estimated from the data (new Figure 7). This strongly supports our idea that slow learning influenced undermatching behavior. We then showed that changes in LMIT were correlated with changes in harvesting efficiency in the data (new Figure 8). This established a link between the bias-variance tradeoff and slow learning, meaning that we confirmed the triad relationships that we predicted by our model in experimental data.

A couple of notes before getting to the substance of this: First, the definition of reward color bias only occurs in methods, and the term is used without explanation in the main text. This was confusing to me.

Our reply:

We apologize for this. We now have redefined various terms for bias for choice and rewards. We defined “color reward imbalance” as the difference between rewards collected for two color targets over trials. This contrasts with “color choice bias”, which is a bias in choice towards one of the color targets. We now clarified this in the text:

(Figure 6 caption) We also estimate session-by-session color reward imbalance that we defined as the imbalance between rewards obtained from two color targets.

(Line 153) Note that there is another well-documented deviation from the matching law [4], which we refer to as color choice bias. This is a bias toward one of the colored targets, and can be quantified as the deviation of the best linear fit to the behavior data from the unity diagonal predicted by the matching law at a reward fraction of 0.5 (see the inner panel of Figure 1 and Methods section). It is important to distinguish two biases: undermatching (slope) and color choice bias (intercept), as these are independent behavioral measures.

Second, as best I understand it, the LMIT correlation measures the ability of a mismatch in number of rewards obtained from each color target to influence bias toward each color many trials later. So just to be clear: its color bias that lags? And the assumption is that reward color bias is averaged over a long enough time period that the colors are approximately equal in value (that is, enough switches in reward rate have taken place)? I believe this is what the authors are attempting to (compactly) explain in ll. 263-265, but that wasn't sufficient for me to follow the argument the first time through.

Our reply:

Again, sorry for the confusion. We now have extensively edited this part to improve clarity:

(line 332) As a direct test of this link between slow reward integration and undermatching, we estimated the effects of very slow reward learning in the experimental data, by measuring long-term reward-choice correlations across, rather than within, experimental sessions.

Here we again do not rely on model-fitting. Instead, as illustrated in Figure 6 our approach is to directly measure the timescale of slow reward integration from the data. For this, we decided to exploit the other type of choice bias in matching behavior, which we refer to as color choice bias. A color choice bias in matching behavior is defined as the intercept, at reward fraction

= 0.5, of a line fit to block-wise matching behavior plotted in choice fraction vs. reward fraction (please see the inner panel in Figure 1b). If animals indeed integrated rewards on a very long timescale, this long-term choice bias should be influenced by imbalance in past reward experience in which the experienced ratio of rewards received from each the two colors deviates from 50% (e.g. see [23]). We can measure slow integration timescale, using session-by-session estimates of color imbalance in rewards and color bias in choice, by asking over how many sessions color reward imbalance influences future color choice bias.

There are two advantages to this approach. One is that we do not rely on any (generative-) model fitting, because the analysis only involves correlation analysis (please see the Methods section for details). The other is that the resulting time constant is a direct measure of the relationship between choice and past rewards. This measure does not suffer from the confounds present in our prior analysis of the time-course of undermatching, in which temporal relationships we observed could have causes other than slow variation in reward history.

To illustrate this model-independent measure, we performed an analysis on simulated data that we generated from the multi-timescale learning model presented in Figure 2. We first estimated the color reward imbalance and the color choice bias for each session of the simulated data. The reward color imbalance is defined as the fraction of reward obtained from one side minus 0.5, while as we previously defined, the color choice bias is the intercept of the matching slope at reward fraction = 0.5. We then take the lagged causal correlations between these two bias vectors (Figure 6). Since the model learns reward history over trials, the color choice bias is supposed to be influenced by the color reward imbalance, with the correlation between the two decaying as the lag increases. We then introduce what we refer to as the longest measurable integration timescale (LMIT), which is the longest time lag for which the correlation is significant. We estimate it by fitting a line to the lagged correlations and taking the intercept of the linear fit at zero correlation (see Methods section for a more formal definition). We then transform this session lag measure to trials by multiplying it with the mean session size so that the LMIT is expressed in trials. The LMIT is a lower bound for the longest timescale of reward integrators.

Now, my main confusion is what, exactly, the authors are trying to establish. Is it simply that monkeys behavior can, in fact, show correlations over such long time scales, arguing that slow integration is plausible? Or is it that the LMIT analysis is somehow proving the correctness of the authors model? The integrator model seems reasonable to me, so Im just missing something about what question the LMIT analysis is supposed to answer.

Related to this: Im perhaps also a bit unclear about what precise point is being made on lines 243-257. I interpreted this (and the following equations) as a demonstration that for very slow integration, the correlation in 4c is equivalent to a bias in the integrator estimates, and so does not provide direct evidence for integration.

But along the same lines, if I consider the next order approximation:

$$I_G^{Slow} = b + \delta$$

$$I_R^{Slow} = b - \delta$$

with $\delta \ll b$ (that is, $b = (I_G^{Slow} + I_R^{Slow})/2$), then the expansion for P_G to next order is

$$P_G \approx \left(1 - \frac{2b}{I_G^{Fast} + I_R^{Fast}}\right) \frac{I_G^{Fast}}{I_G^{Fast} + I_R^{Fast}} + \frac{b + \delta}{I_G^{Fast}} I_G^{Fast} + I_R^{Fast}$$

In this case, doesn't the δ term on the right-hand side work out to a color choice bias?:

$$P_G - \frac{1}{2} - \frac{\delta}{2b} \approx \left(1 - \frac{2b}{I_G^{Fast} + I_R^{Fast}}\right) \left(\frac{I_G^{Fast}}{I_G^{Fast} + I_R^{Fast}} - \frac{1}{2} - \frac{\delta}{2b}\right)$$

That is, if we include a previous reward color bias in b , can't we explain the LMIT effect as well?

Again, I seem to be missing a piece of the logic here.

Our reply:

Thank you very much for this. We realized that the part motivating us to introduce the LMIT analysis was very confusing. Put it simply, each experimental session was too short for fitting models with slow reward integration. We think that this problem may occur again in the future studies, but may be overlooked. So we present our mathematical consideration as one of the main findings of our paper. We greatly expanded our explanation to clarify this point. Please see below the new section that motivates us to introduce an alternative analysis (in which we then introduce the LMIT):

(line 279) We fit the model independently to the data from each session, by maximum likelihood estimation of the relative weights of the three integrators (w_{Fast-1} , w_{Fast-2} , w_{Slow}). We were surprised to find unusually smooth changes in these weights across sessions (Figure 5), as we did not impose any smoothness constraints to our fitting process (i.e. we fitted each session independently; for details see the Methods section). This suggests that the optimization of relative weights – a type of meta-learning – takes place very slowly, and continuously, over sessions.

Further, as we predicted, changes in the weight of the slow integrator are correlated with changes in undermatching (Figure 5). However, due to a potential confound, this correlation is not sufficient to prove the link between undermatching and slow reward integration. This is because the slow reward integrators act as a bias that changes slowly over time, and (as we will show below) any slowly changing bias will lead to correlations with undermatching, regardless of whether that bias is driven by slow reward integration or some other process unrelated to reward history. For example, a bias that is randomly modified from session to session would also show correlations with undermatching, even though that bias does not depend on reward history.

To understand why this is the case, consider again the multi-timescale integrator model with two time constants, one fast and one much slower, as in Figure 2. The fast integrator will generate estimates of the current values of the two alternatives, which we call I_G^{Fast} and I_R^{Fast} respectively. The slow integrator will generate two additional estimates: I_G^{Slow} and I_R^{Slow} , which are almost constant in a block of trials. Then the weighted-averaged income from Green (Red) target can be expressed as

$$I_G = w_{Fast} I_G^{Fast} + w_{Slow} I_G^{Slow},$$

and

$$I_R = w_{Fast} I_R^{Fast} + w_{Slow} I_R^{Slow},$$

respectively. The probability of choosing one of the two alternatives (say G), given by the decision policy following the matching law (Eq.1) is then:

$$P_G = \frac{w_{Fast} I_G^{Fast} + w_{Slow} I_G^{Slow}}{w_{Fast} I_G^{Fast} + w_{Fast} I_R^{Fast} + w_{Slow} I_G^{Slow} + w_{Slow} I_R^{Slow}} \quad (1)$$

When the relative weight of the slow integrators is much smaller than that of the fast integrators, P_G can be rewritten as:

$$P_G \simeq \left(1 - \frac{w_{Slow}}{w_{Fast}} \cdot \frac{I_G^{Slow} + I_R^{Slow}}{I_G^{Fast} + I_R^{Fast}}\right) \frac{I_G^{Fast}}{I_G^{Fast} + I_R^{Fast}} + \frac{w_{Slow}}{w_{Fast}} \frac{I_G^{Slow}}{I_G^{Fast} + I_R^{Fast}},$$

or equivalently

$$P_G - \frac{1}{2} \simeq \left(1 - \frac{w_{Slow}}{w_{Fast}} \cdot \frac{I_{tot}^{Slow}}{I_{tot}^{Fast}}\right) \left[\frac{I_G^{Fast}}{I_G^{Fast} + I_R^{Fast}} - \frac{1}{2}\right] + \frac{w_{Slow}}{2w_{Fast}} \frac{\delta I^{Slow}}{I_{tot}^{Fast}} \quad (2)$$

where we defined the total incomes I_{tot}^{Slow} and I_{tot}^{Fast} to be the sum of the two targets' values estimated on slow and fast timescales, respectively, and defined $\delta I^{Slow} = I_G^{Slow} - I_R^{Slow}$ to be the difference between the two targets' values estimated on the slow timescale.

Equation (2) shows that the tilt of the matching curve relating blockwise choice to blockwise reward fractions (Figure 1b) is given by the first factor in parentheses, and that the slope is less than one. This is because the curve representing matching behavior (mostly undermatching) is obtained by fitting a line to the scatter plot of $P_G - \frac{1}{2}$ against $\frac{I_G^{Fast}}{I_G^{Fast} + I_R^{Fast}} - \frac{1}{2}$ when averaged over a block (which is a relatively fast time scale). Within an experimental session, we can neglect small changes in $I_{G/R}^{Slow}$, as well as changes in block-averaged I_{tot}^{Fast} .

In other words, Equation (2) shows that the degree of undermatching is proportional to $\frac{w_{Slow}}{w_{Fast}} \cdot \frac{I_{tot}^{Slow}}{I_{tot}^{Fast}}$, which in turn is proportional to the relative weight of the slow integrator w_{Slow} itself:

$$1 - \text{Slope} \propto w_{Slow} (I_G^{Slow} + I_R^{Slow}). \quad (3)$$

We thus expect a correlation between the weight of slow integration - w_{Slow} - that we obtain by fitting our model to the data and the average degree of undermatching observed in an experimental session. However, this is not in itself conclusive evidence that undermatching reflects reward history integration over long timescales.

To understand why, consider an alternative model in which reward integration happens only over the fast timescale, but the estimated values of the two targets are the sum of estimates on the fast timescale, $I_{G/R}^{Fast}$ and random subjective-biases B_G and B_R that vary from one experimental session to the next. Note that by "random", we more precisely mean biases to color-targets that are not accounted for by slow reward integration. Thus Eq. (1) is now replaced with

$$P_G = \frac{I_G^{Fast} + B_G}{I_G^{Fast} + I_R^{Fast} + B_G + B_R}, \quad (4)$$

where the relative weight terms are normalized to the bias terms. Again, as long as the biases B_G and B_R are small, similarly to Eq. (2), we obtain

$$P_G - \frac{1}{2} \simeq \left(1 - \frac{B_G + B_R}{I_{tot}^{Fast}}\right) \left[\frac{I_G^{Fast}}{I_G^{Fast} + I_R^{Fast}} - \frac{1}{2}\right] + \frac{B_G - B_R}{2I_{tot}^{Fast}} \quad (5)$$

and in particular

$$1 - \text{Slope} \propto B_G + B_R. \quad (6)$$

Fits of this alternative model would also produce correlations between $B_G + B_R$ and the average undermatching observed in experimental sessions.

Since the slow estimates $I_{G/R}^{\text{Slow}}$ in Eq. (1) are approximately constant over each experimental session (as τ_{Slow} is much longer than the experimental block length), it is difficult to discriminate the slow estimates $I_{G/R}^{\text{Slow}}$ in Eq. (1) from the random bias $B_{G/R}$ in Eq. (5). This means that we cannot discriminate these two models from model-fitting to each experimental session.

In other words, although our model-independent analysis confirmed our prediction of bias-variance tradeoff in the data (Figure 4, direct model fitting to separate experimental sessions is not sufficient to prove our other central prediction, that of a link between slow reward integration and undermatching).

Reviewer #3

In this article, authors investigate learning choices in a probabilistic bandit task in monkeys. Under the assumption that the contingencies are stable, a matching behavior would be optimal. However, here, the contingencies change abruptly, and the monkeys exhibit undermatching. The authors use computational modeling to show that this behavior leads to more rewards for the animals, and that it reflects two time scales of integration of information – a slow time scale reflected by bias in choices, and a fast time scale visible in undermatching (weaker dependence on reward than expected).

This is an interesting and well written paper, and I only have a few concerns. My first concern is in the framing of the paper in terms of optimality, in particular in the introduction. My second concern regards the model-based analysis of the behavioral data.

Our reply:

We appreciate reviewers very positive opinion about our manuscript.

The introduction is framed in terms of optimal behavior, which matching is for the stable version of the task used here. It makes no sense to insist on this so much in the introduction, and to present undermatching as surprisingly unoptimal here, since the task is different: reward probabilities change over time unpredictably. Statements like Paradoxically (l131), and This observation seems to be incompatible with the statement that matching 133 behavior is optimal in this task (l133) are artificial and unhelpful. By definition, the task is different with changes over time, and so it is unsurprising that the optimal behavior is different. Furthermore, this highlights how important assumptions about the knowledge of the task environment are when attempting to define optimal behavior, and thus how hard to interpret any claims of optimality (or suboptimality) in behavior are. In my opinion, this paper would be much better focused on the existence of multiple time scales of integration in learning, rather than on whether this is optimal or not.

Our reply:

Thanks very much for your suggestion. We now have extensively revised the entire Introduction section, in order to shift the focus to the issue of multiple timescales of learning:

(Introduction)

In a changing world, animals have to make decisions that are based on the limited information they can extract from the environment. Models of reinforcement learning and other learning theories assume that subjects' belief about the environment is continuously and gradually updated based on the series of experiences that represent the history of choices and outcomes [46, 41, 63, 62, 57]. In many situations, however, the optimal strategy for updating beliefs and weighing past experiences is not clear. Ideally, past experiences should be weighted according to the degree of uncertainty, or volatility, of the environment [6, 37, 22]: if the environment is stable, it is beneficial to consider a large number of past experiences to better estimate the current state of the environment. If the environment is volatile, it is better to consider only a relatively small number of most recent experiences, as old ones may no longer be informative about the current environment.

Previous studies into flexible reward learning have largely focused on how animals can adjust their learning rate according to the volatility of the environment. Many of these studies have assumed that a single time constant that characterizes the learning rate is continuously modified over time (see e.g.[41, 45, 6, 37, 36]). Real world environments, however, can change on multiple timescales and there are computational reasons to believe that learning on multiple timescales can be highly beneficial [56, 16, 28, 64, 47, 22, 23, 7]. One way to implement such a strategy is to consider multiple learning processes, each operating on a different timescale. For example, if learning involves the integration of choices and outcomes, it could be implemented by considering multiple integrators characterized by different time constants. The final decision would then be determined by a weighted sum of the contributions of the different integrators. To adjust to the volatility of the environment either the timescales of, or the relative weights of, the integrators could be modified. Learning on multiple timescales is difficult to study in laboratory experiments, especially if the timescales range from seconds to months.

Here we analyzed an experiment in which it is possible to identify and characterize integration processes that operate on multiple timescales. We show that these processes cooperate to maximize the performance in a volatile environment in which the theoretically optimal strategy changes periodically without any warning. Interestingly, the subjects in this experiment exhibit a behavior that deviates from the behavior that would be optimal in a stationary environment. Rather than a failure, this deviation reflects a strategy that is adapted to the dynamics of the environment.

More specifically, we analyzed an experiment in which monkeys were trained to perform a dynamic foraging task [55]. In this task, monkeys track the changing reward rates of alternative choices through time. The reward rates changed periodically without any warning. Within each period during which the reward contingencies were kept constant, the probabilistic strategy that maximizes cumulative reward is to follow the so-called matching law, due to a specific reward schedule that is often referred to as the concurrent variable-interval schedule [49, 24].

The matching law constitutes a quantitative description of choice behavior that is often observed in foraging tasks. According to the matching law, subjects distribute their choices across available options in the same proportion as the rewards obtained from those options ([20, 21]). This type of behavior has been observed across a wide range of species including pigeons, rats, monkeys and humans [20, 21, 18, 19, 55, 12, 30, 31, 48, 38, 39]. Although

the matching law provides a simple and elegant description of behavior, actual choice often deviates from the matching.

For example, one common deviation, which is often referred to as undermatching, reveals itself as a more random choice allocation, because the subjects systematically choose less-rewarding options more often than the matching law predicts. Such deviations have been interpreted as a failure on the part of the subjects, reflecting poor discrimination between options [4], or noise in the neural mechanisms underlying decision making [53], or by an imbalance in the learning mechanisms [32]. In our data we also observe significant deviation from the matching law in the form of undermatching; however, we find that overall harvesting performance improves as the behavior deviates more strongly from the matching law.

We hypothesized that this behavioral deviation, and accompanying performance improvement, may reflect an adaptation strategy with multi-timescale learning that optimizes the weighting of both recent and old experiences. Old experiences, when considered, should introduce a bias away from the current reward rates (which are, however, unknown to the subject). This bias will instead reflect information about the long-run average of reward rates. For example, when different choices are equally rewarded in the long-run, the bias will be toward balanced choices, which would make the subjects choose the option with the lower reward rate more often than predicted by the matching law if they knew the true rates. In our experiment, this bias translates into undermatching; choice behavior that appears more random and exploratory than the matching behavior that would be optimal for a subject who knew the true reward contingencies.

However, the bias also has the effect of reducing fluctuations in the subject's estimate of current reward probabilities which must be inferred from finite, stochastic, observations. This reduction in the volatility of estimates compensates for any losses incurred by deviations from strict matching behavior, allowing the overall harvesting performance to increase when more old experiences are taken into account.

To test this hypothesis we estimated the time over which monkeys were integrating the rewards received for each choice, using a simple correlation analysis. We found that this integration-time was much longer than has traditionally been described, varying slowly across days of experiments. More specifically, as predicted, larger deviations from matching behavior correlated with longer integration-times. We also observed that longer integration-times corresponded to less volatile estimates of reward rates. Finally, we also found that the relative contributions of long timescales integrators correlated with the schedule of the experiments (e.g. with the duration of the time intervals between two consecutive experiments), suggesting dynamic adaptation over a long time period including times outside experimental sessions.

My second main concern is about the model-based analysis of the data. I have two sub concerns first, that a number of parameters are hand-chosen, second, that the model fit is done on aggregated rather than raw data (to the best of my understanding of the methods). The authors smooth the choices with arbitrarily chosen kernels (l646-648), then fit a curve to the obtained modified data. Why do the authors not do a trial-by-trial fit of the model with the data, optimizing the log likelihood of choices across the range of relevant parameters, instead? The current procedure likely loses some of the local structure of the data by smoothing, for no clear reason. There are perfectly valid tools for trial by trial analysis of models, and the proposed model is amenable to them (e.g. see multiple didactic papers by Daw and colleagues, Palminteri et al 2017, etc). These methods would have the

advantage of 1) benefitting from all the behavioral richness of choices 2) fitting, rather than choosing key parameters 3) allowing simpler model comparison.

Our reply:

Again, sorry for the confusion. In our previous version of our manuscript, we in fact have fit our model trial-by-trial by using maximum likelihood estimates, as the reviewer suggests. The part that the reviewer mentions is about our model-independent analysis on variance of choice, but we realized that the description was located in a misleading subsection. We apologize for the confusions. We clarified this in the text, by separating into different sections and more detailing explanations in the Methods section

(line 279)

We fit the model independently to the data from each session, by using maximum likelihood estimation of the relative weights of the three integrators (w_{Fast-1} , w_{Fast-2} , w_{Slow}).

(line 751) Model-fitting: We fit three-time-constant model presented in Figure 5 using maximum likelihood estimates. We fit each session independently, except that we set the initial income estimates of the slow integrator as the same as the last income estimates of the slow integrator, since the slow integrator may carry over the previous estimates of the income. We, however, found that this assumption is not critical, due to the confound discussed in the manuscript. We set the time constants as $\tau_{Fast-1} = 2$ trials, $\tau_{Fast-2} = 20$ trials, and $\tau_{Slow} = 1000$ trials, though the exact choice of these do not affect our results.

(line 758) Estimating variance of choice in the data: We computed the variance of Monkey's choice in Figure 3 as follows. First, monkey's choice time series was smoothed via two half-gaussian kernels with standard deviations of $\sigma = 8$ trials and $\sigma = 50$ trials, with a span of 200 trials. This gave us two time-series, a fast one with $\sigma = 8$ and a slow one with $\sigma = 50$. We computed the variance of the fast one over the slow one. We showed squared root of variance in figures for an illustrative purpose.

Also we found the suggested citation (Palminteri et al. 2017) very useful for us, since we also found our model-fitting inconclusive. We now have cited the article in our manuscript. Thank you very much.

Bibliography

- [1] L. Acerbi, S. Vijayakumar, and D. M. Wolpert. On the origins of suboptimality in human probabilistic inference. *PLoS Comput Biol*, 10(6):e1003661, 2014.
- [2] R. P. Adams and D. J. MacKay. Bayesian online changepoint detection. *arXiv preprint arXiv:0710.3742*, 2007.
- [3] C. F. Aparicio and W. M. Baum. Dynamics of choice: relative rate and amount affect local preference at three different time scales. *J Exp Anal Behav*, 91(3):293–317, May 2009.
- [4] W. M. Baum. On two types of deviation from the matching law: bias and undermatching. *J Exp Anal Behav*, 22(1):231–242, 1974.
- [5] J. M. Beck, W. J. Ma, X. Pitkow, P. E. Latham, and A. Pouget. Not noisy, just wrong: the role of suboptimal inference in behavioral variability. *Neuron*, 74(1):30–39, 2012.
- [6] T. E. Behrens, M. W. Woolrich, M. E. Walton, and M. F. Rushworth. Learning the value of information in an uncertain world. *Nat. Neurosci.*, 10(9):1214–1221, Sep 2007.
- [7] M. K. Benna and S. Fusi. Computational principles of synaptic memory consolidation. *Nature neuroscience*, 2016.
- [8] A. Bernacchia, H. Seo, D. Lee, and X.-J. Wang. A reservoir of time constants for memory traces in cortical neurons. *Nature neuroscience*, 14(3):366–372, 2011.
- [9] E. S. Bromberg-Martin, M. Matsumoto, H. Nakahara, and O. Hikosaka. Multiple timescales of memory in lateral habenula and dopamine neurons. *Neuron*, 67(3):499–510, 2010.
- [10] J. D. Cohen, S. M. McClure, and J. Y. Angela. Should i stay or should i go? how the human brain manages the trade-off between exploitation and exploration. *Philosophical Transactions of the Royal Society of London B: Biological Sciences*, 362(1481):933–942, 2007.
- [11] J. Y. Cohen, M. W. Amoroso, and N. Uchida. Serotonergic neurons signal reward and punishment on multiple timescales. *Elife*, 4:e06346, 2015.
- [12] G. S. Corrado, L. P. Sugrue, H. S. Seung, and W. T. Newsome. Linear-nonlinear-poisson models of primate choice dynamics. *J Exp Anal Behav*, 84(3):581–617, 2005.
- [13] O. Dan, D. Hochner-Celnikier, A. Solnica, and Y. Loewenstein. Association of catastrophic neonatal outcomes with increased rate of subsequent cesarean deliveries. *Obstetrics & Gynecology*, 129(4):671–675, 2017.

- [14] P. Dayan and T. J. Sejnowski. Exploration bonuses and dual control. *Machine Learning*, 25(1):5–22, 1996.
- [15] J. Friedman, T. Hastie, and R. Tibshirani. *The elements of statistical learning*, volume 1. Springer series in statistics Springer, Berlin, 2001.
- [16] S. Fusi, W. F. Asaad, E. K. Miller, and X. J. Wang. A neural circuit model of flexible sensorimotor mapping: learning and forgetting on multiple timescales. *Neuron*, 54:319–333, Apr 2007.
- [17] S. Fusi, P. J. Drew, and L. F. Abbott. Cascade models of synaptically stored memories. *Neuron*, 45(4):599–611, Feb 2005.
- [18] C. R. Gallistel. Foraging for brain stimulation: toward a neurobiology of computation. *Cognition*, 50:151–170, 1994.
- [19] C. R. Gallistel, T. A. Mark, A. P. King, and P. E. Latham. The rat approximates an ideal detector of changes in rates of reward: implications for the law of effect. *J Exp Psychol Anim Behav Process*, 27:354–372, Oct 2001.
- [20] R. J. Herrnstein. Relative and absolute strength of response as a function of frequency of reinforcement. *J. Exp. Anal. Behav.*, 4:267–272, 1961.
- [21] L. D. Herrnstein RJ, Rachlin H. The matching law: papers in psychology and economics. *Harvard University Press*, 1997.
- [22] K. Iigaya. Adaptive learning and decision-making under uncertainty by metaplastic synapses guided by a surprise detection system. *Elife*, 5:e18073, 2016.
- [23] K. Iigaya, M. S. Fonseca, M. Murakami, Z. F. Mainen, and P. Dayan. The long and the short of serotonergic stimulation: Optogenetic activation of dorsal raphe serotonergic neurons changes the learning rate for rewards. *bioRxiv*, page 215400, 2017.
- [24] K. Iigaya and S. Fusi. Dynamical regimes in neural network models of matching behavior. *Neural computation*, 25:1–20, 2013.
- [25] K. Iigaya, L. Sugrue, G. Corrado, W. Newsome, and S. Fusi. Deviations from the matching law reflect reward integration over multiple timescales. *Cosyne Abstract*, 2013.
- [26] H. F. Kim, A. Ghazizadeh, and O. Hikosaka. Dopamine neurons encoding long-term memory of object value for habitual behavior. *Cell*, 163(5):1165–1175, 2015.
- [27] H. F. Kim and O. Hikosaka. Distinct Basal Ganglia Circuits Controlling Behaviors Guided by Flexible and Stable Values. *Neuron*, Aug 2013.
- [28] K. P. Kording, J. B. Tenenbaum, and R. Shadmehr. The dynamics of memory as a consequence of optimal adaptation to a changing body. *Nature neuroscience*, 10(6):779–786, 2007.
- [29] G. La Camera, A. Rauch, D. Thurbon, H.-R. Lüscher, W. Senn, and S. Fusi. Multiple time scales of temporal response in pyramidal and fast spiking cortical neurons. *Journal of neurophysiology*, 96(6):3448–3464, 2006.

- [30] B. Lau and P. W. Glimcher. Dynamic response-by-response models of matching behavior in rhesus monkeys. *J Exp Anal Behav*, 84(3):555–579, 2005.
- [31] B. Lau and P. W. Glimcher. Value representations in the primate striatum during matching behavior. *Neuron*, 58(3):451–463, 2008.
- [32] Y. Loewenstein. Robustness of learning that is based on covariance-driven synaptic plasticity. *PLoS Comput. Biol.*, 4(3):e1000007, Mar 2008.
- [33] Y. Loewenstein, D. Prelec, and H. S. Seung. Operant matching as a Nash equilibrium of an intertemporal game. *Neural Comput*, 21:2755–2773, Oct 2009.
- [34] Y. Loewenstein and H. S. Seung. Operant matching is a generic outcome of synaptic plasticity based on the covariance between reward and neural activity. *Proc. Natl. Acad. Sci. U.S.A.*, 103:15224–15229, Oct 2006.
- [35] B. N. Lundstrom, A. L. Fairhall, and M. Maravall. Multiple timescale encoding of slowly varying whisker stimulus envelope in cortical and thalamic neurons in vivo. *Journal of Neuroscience*, 30(14):5071–5077, 2010.
- [36] M. R. Nassar, K. M. Rumsey, R. C. Wilson, K. Parikh, B. Heasley, and J. I. Gold. Rational regulation of learning dynamics by pupil-linked arousal systems. *Nat. Neurosci.*, 15(7):1040–1046, Jul 2012.
- [37] M. R. Nassar, R. C. Wilson, B. Heasley, and J. I. Gold. An approximately Bayesian delta-rule model explains the dynamics of belief updating in a changing environment. *J. Neurosci.*, 30(37):12366–12378, Sep 2010.
- [38] T. Neiman and Y. Loewenstein. Reinforcement learning in professional basketball players. *Nature communications*, 2:569, 2011.
- [39] T. Neiman and Y. Loewenstein. Covariance-based synaptic plasticity in an attractor network model accounts for fast adaptation in free operant learning. *Journal of Neuroscience*, 33(4):1521–1534, 2013.
- [40] Y. Niv, N. D. Daw, D. Joel, and P. Dayan. Tonic dopamine: opportunity costs and the control of response vigor. *Psychopharmacology*, 191(3):507–520, 2007.
- [41] J. M. Pearce and G. Hall. A model for pavlovian learning: variations in the effectiveness of conditioned but not of unconditioned stimuli. *Psychological review*, 87(6):532, 1980.
- [42] M. L. Platt and P. W. Glimcher. Neural correlates of decision variables in parietal cortex. *Nature*, 400:233–238, Jul 1999.
- [43] C. Pozzorini, R. Naud, S. Mensi, and W. Gerstner. Temporal whitening by power-law adaptation in neocortical neurons. *Nature neuroscience*, 16(7):942–948, 2013.
- [44] D. Prelec. Matching, maximizing, and the hyperbolic reinforcement feedback function. *Psychological Review*, 89(3):189, 1982.
- [45] K. Preuschoff and P. Bossaerts. Adding prediction risk to the theory of reward learning. *Annals of the New York Academy of Sciences*, 1104(1):135–146, 2007.

- [46] R. A. Rescorla, A. R. Wagner, et al. A theory of pavlovian conditioning: Variations in the effectiveness of reinforcement and nonreinforcement. *Classical conditioning II: Current research and theory*, 2:64–99, 1972.
- [47] A. Roxin and S. Fusi. Efficient partitioning of memory systems and its importance for memory consolidation. *PLoS Comput. Biol.*, 9(7):e1003146, Jul 2013.
- [48] R. B. Rutledge, S. C. Lazzaro, B. Lau, C. E. Myers, M. A. Gluck, and P. W. Glimcher. Dopaminergic drugs modulate learning rates and perseveration in Parkinson’s patients in a dynamic foraging task. *J. Neurosci.*, 29:15104–15114, Dec 2009.
- [49] Y. Sakai and T. Fukai. The actor-critic learning is behind the matching law: matching versus optimal behaviors. *Neural Comput*, 20(1):227–251, 2008.
- [50] Y. Sakai and T. Fukai. When does reward maximization lead to matching law? *PLoS ONE*, 3:e3795, 2008.
- [51] W. Schultz, P. Dayan, and P. R. Montague. A neural substrate of prediction and reward. *Science*, 275(5306):1593–1599, Mar 1997.
- [52] H. S. Seung. Learning in spiking neural networks by reinforcement of stochastic synaptic transmission. *Neuron*, 40:1063–1073, Dec 2003.
- [53] A. Soltani, D. Lee, and X. J. Wang. Neural mechanism for stochastic behaviour during a competitive game. *Neural Networks*, 19(8):1075 – 1090, 2006.
- [54] A. Soltani and X.-J. Wang. A Biophysically Based Neural Model of Matching Law Behavior: Melioration by Stochastic Synapses. *J. Neurosci.*, 26(14):3731–3744, 2006.
- [55] L. P. Sugrue, G. S. Corrado, and W. T. Newsome. Matching Behavior and the Representation of Value in the Parietal Cortex. *Science*, 304(5678):1782–1787, 2004.
- [56] R. S. Sutton. Td models: Modeling the world at a mixture of time scales. In *ICML*, volume 12, pages 531–539. Citeseer, 1995.
- [57] R. S. Sutton and A. G. Barto. *Reinforcement Learning: An Introduction (Adaptive Computation and Machine Learning)*. A Bradford Book, Mar. 1998.
- [58] N. Ulanovsky, L. Las, D. Farkas, and I. Nelken. Multiple time scales of adaptation in auditory cortex neurons. *Journal of Neuroscience*, 24(46):10440–10453, 2004.
- [59] I. Vilares, J. D. Howard, H. L. Fernandes, J. A. Gottfried, and K. P. Kording. Differential representations of prior and likelihood uncertainty in the human brain. *Current Biology*, 22(18):1641–1648, 2012.
- [60] X. J. Wang. Probabilistic decision making by slow reverberation in cortical circuits. *Neuron*, 36(5):955–968, Dec 2002.
- [61] B. Wark, A. Fairhall, and F. Rieke. Timescales of inference in visual adaptation. *Neuron*, 61(5):750–761, 2009.
- [62] C. J. Watkins and P. Dayan. Q-learning. *Machine learning*, 8(3-4):279–292, 1992.

- [63] C. J. C. H. Watkins. *Learning from delayed rewards*. PhD thesis, King's College, Cambridge, 1989.
- [64] R. C. Wilson, M. R. Nassar, and J. I. Gold. A mixture of delta-rules approximation to bayesian inference in change-point problems. *PLoS Comput. Biol.*, 9(7):e1003150, Jul 2013.
- [65] M. K. Wittmann, N. Kolling, R. Akaishi, B. K. Chau, J. W. Brown, N. Nelissen, and M. F. Rushworth. Predictive decision making driven by multiple time-linked reward representations in the anterior cingulate cortex. *Nature Communications*, 7, 2016.

Reviewers' Comments:

Reviewer #1:

Remarks to the Author:

In the revised manuscript, the authors have addressed all of my concerns. Both achievements and limitations of the present data analysis and model have been much better clarified compared to the previous manuscript. This reviewer does not have further comments except for a minor issue. Figure 6a is hard to see and should be shown with larger fonts.

Reviewer #2:

Remarks to the Author:

First of all, I would like to congratulate the authors on a significantly clarified manuscript. The claims and their relationships are now much clearer, which I believe greatly helps the appeal of the work. I also appreciate their thoughtful responses to my critiques.

However, while I remain positively disposed to the manuscript, several aspects of the presentation now seem to me unnecessarily muddled, and I can now see clearly at least one significant concern. I believe these can be addressed, but I would like to see that done before I can endorse publication.

First, the presentation. While this is much clearer than before, I find the paper still quite confusing in the relationships among its claims. It's possible this is just my confusion, but my understanding of the key results is that:

1. A bias-variance tradeoff can explain undermatching as observed
2. Integration over multiple time scales leads to a bias-variance tradeoff

Importantly, both of these effects together are sufficient but not necessary for the observed behavior. What I believe is confusing to the reader is that claims 1 and 2 are intermixed in Section 2.2, section 2.3 is about establishing that the bias-variance tradeoff determines optimal parameter values in a toy model, and Section 2.5 is about attempting to use an independent check to argue (3) that a separate effect (color bias) can also be explained by integration, and so (by parsimony?) the combination of 1+2 and 3 is evidence for integration on multiple time scales.

Some suggestions that could help clarify this:

- Section 2.3: As I read it, the purpose of this section is to show that, **in a toy problem** the bias-variance tradeoff for the **inference problem** for the reward probabilities of a two-armed bandit can be used to select an optimal value of w_{slow} if the underlying estimation method is integration over multiple time scales. There's nothing wrong with this, and it's fine to build intuition that bias-variance tradeoffs can set optimal parameter values, but I had two issues with reading this:

o I don't think it's made clear what readers are supposed to get out of this. I've given my take above, but this may not be what the authors intended. I think it can be valuable to use a toy model to build readers' intuition, but if this is the desired outcome, it could be more clearly labeled as such.

o My bigger issue is not even that the toy problem reward schedule, a bandit task, is not the same as the problem the monkeys face, concurrent VI schedules. Real problems aren't always analytically tractable. My difficulty is with the fact that there is a subtle elision going on here with the term variance: In the toy problem, the variance is in the estimator of the block reward probability. This is **variance in probability estimation**. By contrast, the quantity called variance in Figure 3d is the variance of something else entirely. The relationship between these two variances is wholly unclear to me:

♣ First of all, because the problem the monkeys are solving is not an inference problem, as in the toy model, but a reward optimization problem, which is illustrated in 3e. Now, in general, bias-variance tradeoffs help set parameters, but can't this be argued on general grounds if this is all that's needed? What are readers supposed to take away from a toy problem that has not only a different reward schedule, but a different optimization criterion entirely? It would help for the authors to clarify this.

♣ Second, I am confused by the authors' "variance" metric for "choice probability." I could understand smoothing to estimate the time-varying probability of monkeys' choices, and I could understand a bias-variance tradeoff involved with estimating this correctly, but I don't understand the division by a slower moving average (I'm assuming from the text that variance is calculated on the ratio, and is not the ratio of the variances). Nor do I understand why this inference problem is the one we should be concerned with, since again, this is not the problem the monkeys are solving. Why should readers care about the tradeoff between bias and variance in **estimating the probabilities of monkeys' choices**?

- Section 2.5/Figure 5: The whole point here is that integration over multiple time scales is sufficient but not necessary for undermatching, right? Again, signposting this up front would help.

- I very much appreciate Figure 6a. This helps clarify the authors' claims somewhat. Clearly, the main claim of the paper is that the connection between integration over multiple time scales and undermatching is two-way. Not only is slow integration sufficient, it is also implied by undermatching. However, this figure raises two issues:

o I'm not sure how to get from undermatching to slow integration via the arrows in the figure. My reading of 2.5.1 is that the reasoning is slow integration => LMIT, but I don't see LMIT => slow integration established. The strongest argument, as I understand it, is that slow integration implies both undermatching AND color bias, and these are independent effects, so it's a parsimonious explanation. I may have missed something, though.

o Second, and **this is a potentially serious flaw**: I missed this on a previous reading, but of course, color bias and undermatching are **not** independent. They are independent conceptually, and they are independent if one considers the distribution of all lines in the plane, but the **statistical estimates** of slope and intercept **are** correlated, with correlation matrix $\sigma^2 (X^{\text{top}} X)^{-1}$. It is important that the authors demonstrate that the off-diagonal elements of this matrix are small, since their argument hinges on the claim that estimates of these values from the data are independent. Otherwise, the LMIT analysis does not offer an independent check.

Small things:

- Sec 2.2 title: "Changes in matching behavior are predicted by..." "Changes in matching behavior are could be explained by..."?

- Figure 2 caption: I don't see anything in this figure related to performance. Clearly integration on multiple time scales, if true, could lead to undermatching, as shown, but there's nothing here about reward harvesting.

- If you're not plotting variance, please fix the captions and figure legends. Variance is perfectly fine.

- Fig 6b: axis label should be color imbalance?

- Fig 9: With the exception of panel b, none of these are well described by a linear regression. You can do the regression, of course, but a correlation simply doesn't capture what's going on in the raw data. Panel d is pretty much straight out of Anscombe's quartet.

Reviewer #3:

Remarks to the Author:

I am satisfied by the thorough revision done by the authors.

Reviewer #2

First of all, I would like to congratulate the authors on a significantly clarified manuscript. The claims and their relationships are now much clearer, which I believe greatly helps the appeal of the work. I also appreciate their thoughtful responses to my critiques. However, while I remain positively disposed to the manuscript, several aspects of the presentation now seem to me unnecessarily muddled, and I can now see clearly at least one significant concern. I believe these can be addressed, but I would like to see that done before I can endorse publication.

Our reply:

Thanks for the positive comments. We took seriously all the comments below and we hope that we addressed all the remaining concerns.

First, the presentation. While this is much clearer than before, I find the paper still quite confusing in the relationships among its claims. Its possible this is just my confusion, but my understanding of the key results is that: 1. A bias-variance tradeoff can explain undermatching as observed 2. Integration over multiple time scales leads to a bias-variance tradeoff. Importantly, both of these effects together are sufficient but not necessary for the observed behavior.

What I believe is confusing to the reader is that claims 1 and 2 are intermixed in Section 2.2, section 2.3 is about establishing that the bias-variance tradeoff determines optimal parameter values in a toy model, and Section 2.5 is about attempting to use an independent check to argue (3) that a separate effect (color bias) can also be explained by integration, and so (by parsimony?) the combination of 1+2 and 3 is evidence for integration on multiple time scales.

Our reply:

We agree that we clearly needed to further clarify the main claims of our manuscript. The three claims the Reviewer refers to would probably be less confusing if reformulated as in the following list, where we summarize the logical steps of the article:

- **Observation (data; Figure 1): we observe undermatching (already observed), and undermatching changes over time (new observation) in monkey behavioral data.**
- **Claim 1 (theory; Figure 2): Such changes in undermatching behavior can be accounted for by models in which reward learning is integrated over multiple timescales. In particular, in such models undermatching is a consequence of reward integration over a very long timescale (i.e. much longer than a typical block of trials with fixed reward contingencies).**
- **Claim 2 (theory; Figure 3): A generic bias-variance tradeoff arises in a wide range of inference tasks (see also Eq.17 in Supplementary material). Our analysis of a toy problem shows that a model like the one of Claim 1 (i.e. that integrates rewards over multiple timescales) can deal with this tradeoff (Figure 3a,b,c). Importantly, we show with model simulations using the same reward schedule as the actual monkey experiments (we added some new results, as suggested by the Reviewer; please see below) that the results in the toy problem can be easily extended to describe the monkey experiments (Figure 3d,e) and would have similar computational properties. Hence the predictions about the role of integration on long timescales extend to the data.**

- **Testing our claims in data in a model-independent way: is there evidence of a bias-variance tradeoff in the monkey behavior, and did monkeys actually use reward integration over a very long timescale to address the bias-variance tradeoff? (data; Figures 4-10)**
 - **Figure 4: Predicted bias-variance trade-off indeed exists in our behavioral data, though we do not know if monkeys used reward integration over a very long timescale to deal with this problem.**
 - **Figure 5: Fitting our simple computational model to data supports slow learning, but it is not sufficient.**
 - **Figure 6: Introduction of an alternative approach, in which we show more directly (i.e. without fitting a model to the data) that reward is integrated over long timescales.**
 - **Figure 7: We confirm the evidence of reward integration over a very long timescale in the experimental data. Changes in the degree of undermatching is indeed correlated with the extremely slow reward history integration, consistent with our prediction that undermatching is accounted for by reward integration over a very long timescale .**
 - **Figure 8: Changes in reward harvesting performance is also correlated with very slow reward history integration, consistent with our prediction that monkeys used reward integration over a very long timescale to resolve the bias-variance trade-off.**
 - **Figure 9: Changes in reward integration over a very long timescale appear to be influenced by the experimental schedule over multiple days.**
 - **Figure 10: Changes in reward integration over a very long timescale can be accounted for by a previously proposed synaptic meta-plasticity model.**

We now have added the following paragraph to the Introduction:

To summarize, in what follows we will show that 1. the degree of undermatching changes over time in monkey behavior. 2. Such changes in undermatching are predicted by a simple model in which reward is integrated over multiple timescales (short and long ones). 3. This simple toy model can address a bias-variance trade-off that subjects always face in volatile environments. 4. We found evidence of the bias-variance trade-off in monkey behavior. 5. We then show that monkeys also used multiple timescales of reward integration to tackle this tradeoff, and in particular, that the degree of undermatching is correlated with an integration of reward history over long timescales. This slow reward history integration is correlated with a measure of behavioral performance.

Some suggestions that could help clarify this:- Section 2.3: As I read it, the purpose of this section is to show that, in a toy problem the bias-variance tradeoff for the inference problem for the reward probabilities of a two-armed bandit can be used to select an optimal value of w_{slow} if the underlying estimation method is integration over multiple time scales. There's nothing wrong with this, and it's fine to build intuition that bias-variance tradeoffs can set optimal parameter values, but I had two issues with reading this: I don't think it's made clear what readers are supposed to get out of this. I've given my take above, but this may not be what the authors intended. I think it can be valuable to use a toy model to build readers' intuition, but if this is the desired outcome, it could be more clearly labeled as such. My bigger issue is not even that the toy problem reward schedule, a bandit task, is not

the same as the problem the monkeys face, concurrent VI schedules. Real problems aren't always analytically tractable. My difficulty is with the fact that there is a subtle elision going on here with the term variance: In the toy problem, the variance is in the estimator of the block reward probability. This is variance in probability estimation. By contrast, the quantity called variance in Figure 3d is the variance of something else entirely. The relationship between these two variances is wholly unclear to me: First of all, because the problem the monkeys are solving is not an inference problem, as in the toy model, but a reward optimization problem, which is illustrated in 3e. Now, in general, bias-variance tradeoffs help set parameters, but can't this be argued on general grounds if this is all that's needed? What are readers supposed to take away from a toy problem that has not only a different reward schedule, but a different optimization criterion entirely? It would help for the authors to clarify this.

Our reply:

We thank the reviewer for this comment and the suggestions. We started with the toy problem for two reasons: 1. it gives an intuition about the computational advantage of integration over multiple timescales, and how undermatching can arise from this computational process of integration; 2. it can be solved analytically.

However, as the reviewer rightly pointed out, the toy problem is not exactly the same as the problem solved by the monkeys in the experiment that we studied. For this reason, we have simulated our model using the actual paradigm of the monkey experiment to show that the intuition obtained from the toy model can be applied to the actual experiment (Figure 3de). This result is not too surprising because of the generality of the bias-variance tradeoff (see the discussion above). However, the Reviewer is actually right that there was still one missing logical step in the previous version of the manuscript.

More specifically, we agree that variance in the subject's reward rate estimation and the variance in subject's choice could be different in general. However, at least in our setup, they should be strongly correlated because the local matching law (Eq.(1)) predicts a linear relationship between choice probability and the reward rate estimation (if the total reward rate is relatively constant), and this local-matching has been previously validated in our data (e.g. in [55]).

We now have explicitly tested by model simulations in the same experimental schedule as data that the variance in choice (generated by our local-matching model) and the variance in the reward rate (estimated by our same local-matching model) are significantly correlated (Please see new Figure S2). We now discuss this issue in the text:

(line 267)

We now show how our analytical model's predictions can be applied to the actual experimental task. We first note that the variance of inference is related to the variance of choice in our experiment. To see it, we simulated our model with a single learning rate (the same as the one of the previously validated model [55]). We used the parameters that were estimated by model fitting (maximum likelihood) for each session of the experiments for Monkey F. Figure S2 clearly shows a positive correlation between the variance of inferred reward rate and the variance of choice behavior.

Therefore the bias-variance tradeoff in the inference, which we showed using analytical calculations, translates into an analogous tradeoff that should be observable in subject's choice behavior in the experiment.

Second, I am confused by the authors variance metric for choice probability. I could understand smoothing to estimate the time-varying probability of monkeys choices, and I could understand a bias-variance tradeoff involved with estimating this correctly, but I dont understand the division by a slower moving average (Im assuming from the text that variance is calculated on the ratio, and is not the ratio of the variances). Nor do I understand why this inference problem is the one we should be concerned with, since again, this is not the problem the monkeys are solving. Why should readers care about the tradeoff between bias and variance in estimating the probabilities of monkeys choices?

Our reply:

As we discussed in the previous point, the subject's choice probability and the subject's estimated reward rates are linearly correlated in the local matching policy that subjects used in this task. Thus we can observe the bias-variance tradeoff of subjects in their choice behavior. For estimating the variance of choice, we took the average distance between two low-passed signals (the low-passed signal using the smaller sigma filter represents a local estimate of probability of choice, the one with the larger sigma represents the block-scale average of this local estimate, and thus the mean squared distance between the two measures the variance of the local estimate). Therefore this does not involve any division.

We also note that the bias-variance tradeoff of monkeys (subjects) that we discussed in the manuscript is not related to the bias-variance tradeoff of humans (observers of the subjects) in estimating the subjects' probability of choice. We did not mention the latter in our manuscript.

(Methods section)

We computed the variance of Monkey's choice in Figures 3 and S2 as follows. First, monkey's choice time series was smoothed via two half-gaussian kernels with standard deviations of $\sigma = 8$ trials and $\sigma = 50$ trials, with a span of 200 trials. This gave us two time-series, a fast one with $\sigma = 8$ and a slow one with $\sigma = 50$. The faster signal represents a local estimate of probability of choice, the slower one represents the block-scale average of this local estimate. We then estimated the variance of the local estimate as the average squared distance between the fast time series and the slow time series. We showed squared root of variance in Figure 3 for illustrative purpose.

We estimated the variance of inference in Figure S2 in the same fashion, using the income from the Green target in the model.

Section 2.5/Figure 5: The whole point here is that integration over multiple time scales is sufficient but not necessary for undermatching, right? Again, signposting this up front would help.

Our reply:

The Reviewer is right. As we described in our response to the reviewer's first point, fitting our simple computational model to data supports slow learning, but it is not sufficient.

I very much appreciate Figure 6a. This helps clarify the authors claims somewhat. Clearly, the main claim of the paper is that the connection between integration over multiple time scales and undermatching is two-way. Not only is slow integration sufficient, it is also implied by undermatching. However, this figure raises two issues: I'm not sure how to get from undermatching to slow integration via the arrows in the figure. My reading of 2.5.1 is that the reasoning is slow integration \Rightarrow LMIT, but I don't see LMIT \Rightarrow slow integration established. The strongest argument, as I understand it, is that slow integration implies both undermatching AND color bias, and these are independent effects, so it's a parsimonious explanation. I may have missed something, though.

Our reply:

Thanks for your comment, it made us realize that the arrows could be misinterpreted. In our analysis, we estimated how reward imbalance history can predict future color choice bias. That is the longest measurable integration timescale (LMIT). The arrows in the previous version of the figure did not mean any directionality or causality. All they indicated was the existence of correlations, not necessarily causality. We now have modified accordingly the figure.

Second, and **this is a potentially serious flaw**: I missed this on a previous reading, but of course, color bias and undermatching are **not** independent. They are independent conceptually, and they are independent if one considers the distribution of all lines in the plane, but the **statistical estimates** of slope and intercept **are** correlated, with correlation matrix $\sigma^2(X^T X)^{-1}$. It is important that the authors demonstrate that the off-diagonal elements of this matrix are small, since their argument hinges on the claim that estimates of these values from the data are independent. Otherwise, the LMIT analysis does not offer an independent check.

Our reply:

This is a legitimate concern, though it is difficult to predict how the correlations between slopes and intercepts affects the LMIT, which is a non-linear function of choice color bias and reward balance history. As suggested by the Reviewer, we computed the correlation between slopes and intercepts. We found that the correlation was not significant but very weak ($p > 0.1$, $r = -0.07$). We thus think that the LMIT analysis offers an independent check.

We now have mentioned this in the main text:

(line 409) We note that, in general, measurements of a slope and an intercept in a linear regression can be correlated; however, we did not see a significant correlation in our analysis.

Small things:- Sec 2.2 title: Changes in matching behavior are predicted by Changes in matching behavior are could be explained by?

Our reply:

We have changed the title accordingly.

Figure 2 caption: I dont see anything in this figure related to performance. Clearly integration on multiple time scales, if true, could lead to undermatching, as shown, but theres nothing here about reward harvesting.- If youre not plotting variance, please fix the captions and figure legends. Variance is perfectly fine.

Our reply:

We now have changed the caption accordingly.

Fig 6b: axis label should be color imbalance?

Our reply:

Yes, thanks for spotting the mistake. We now have changed the label.

Fig 9: With the exception of panel b, none of these are well described by a linear regression. You can do the regression, of course, but a correlation simply doesnt capture whats going on in the raw data. Panel d is pretty much straight out of Anscombes quartet.

Our reply:

We agree that the data is rich and not reducible to a linear relationship. The data points in figure 9 probably reflect many, complex, aspects of the lives of the monkeys over the many months during which the data was collected. However, we have no control over what monkeys were doing between experiments. So we cannot fully account for the observed variations. The primary goal of Figure 9 was to report the existence of correlations between the LMIT and the duration of inter-experiment-interval, or the experimental sessions, which we can account for by our model of metaplastic synapses [17, 22], as a form of memory consolidation and forgetting, though it is just one of the many aspects that characterize what happens between consecutive experiments. We appreciate the comment and we now have changed the manuscript as follows:

(line 438)

Note that we are aware that linear regression cannot fully capture the richness of the data. It is only a simple way of characterizing what happens in the intervals between experimental sessions (see Figure 9. However, the correlations that we report here can be interpreted as a consequence the theoretical framework of metaplastic synapses [17, 22] that we discuss in the Discussions.

Bibliography

- [1] L. Acerbi, S. Vijayakumar, and D. M. Wolpert. On the origins of suboptimality in human probabilistic inference. *PLoS Comput Biol*, 10(6):e1003661, 2014.
- [2] R. P. Adams and D. J. MacKay. Bayesian online changepoint detection. *arXiv preprint arXiv:0710.3742*, 2007.
- [3] C. F. Aparicio and W. M. Baum. Dynamics of choice: relative rate and amount affect local preference at three different time scales. *J Exp Anal Behav*, 91(3):293–317, May 2009.
- [4] W. M. Baum. On two types of deviation from the matching law: bias and undermatching. *J Exp Anal Behav*, 22(1):231–242, 1974.
- [5] J. M. Beck, W. J. Ma, X. Pitkow, P. E. Latham, and A. Pouget. Not noisy, just wrong: the role of suboptimal inference in behavioral variability. *Neuron*, 74(1):30–39, 2012.
- [6] T. E. Behrens, M. W. Woolrich, M. E. Walton, and M. F. Rushworth. Learning the value of information in an uncertain world. *Nat. Neurosci.*, 10(9):1214–1221, Sep 2007.
- [7] M. K. Benna and S. Fusi. Computational principles of synaptic memory consolidation. *Nature neuroscience*, 2016.
- [8] A. Bernacchia, H. Seo, D. Lee, and X.-J. Wang. A reservoir of time constants for memory traces in cortical neurons. *Nature neuroscience*, 14(3):366–372, 2011.
- [9] E. S. Bromberg-Martin, M. Matsumoto, H. Nakahara, and O. Hikosaka. Multiple timescales of memory in lateral habenula and dopamine neurons. *Neuron*, 67(3):499–510, 2010.
- [10] J. D. Cohen, S. M. McClure, and J. Y. Angela. Should i stay or should i go? how the human brain manages the trade-off between exploitation and exploration. *Philosophical Transactions of the Royal Society of London B: Biological Sciences*, 362(1481):933–942, 2007.
- [11] J. Y. Cohen, M. W. Amoroso, and N. Uchida. Serotonergic neurons signal reward and punishment on multiple timescales. *Elife*, 4:e06346, 2015.
- [12] G. S. Corrado, L. P. Sugrue, H. S. Seung, and W. T. Newsome. Linear-nonlinear-poisson models of primate choice dynamics. *J Exp Anal Behav*, 84(3):581–617, 2005.
- [13] O. Dan, D. Hochner-Celnikier, A. Solnica, and Y. Loewenstein. Association of catastrophic neonatal outcomes with increased rate of subsequent cesarean deliveries. *Obstetrics & Gynecology*, 129(4):671–675, 2017.

- [14] P. Dayan and T. J. Sejnowski. Exploration bonuses and dual control. *Machine Learning*, 25(1):5–22, 1996.
- [15] J. Friedman, T. Hastie, and R. Tibshirani. *The elements of statistical learning*, volume 1. Springer series in statistics Springer, Berlin, 2001.
- [16] S. Fusi, W. F. Asaad, E. K. Miller, and X. J. Wang. A neural circuit model of flexible sensorimotor mapping: learning and forgetting on multiple timescales. *Neuron*, 54:319–333, Apr 2007.
- [17] S. Fusi, P. J. Drew, and L. F. Abbott. Cascade models of synaptically stored memories. *Neuron*, 45(4):599–611, Feb 2005.
- [18] C. R. Gallistel. Foraging for brain stimulation: toward a neurobiology of computation. *Cognition*, 50:151–170, 1994.
- [19] C. R. Gallistel, T. A. Mark, A. P. King, and P. E. Latham. The rat approximates an ideal detector of changes in rates of reward: implications for the law of effect. *J Exp Psychol Anim Behav Process*, 27:354–372, Oct 2001.
- [20] R. J. Herrnstein. Relative and absolute strength of response as a function of frequency of reinforcement. *J. Exp. Anal. Behav.*, 4:267–272, 1961.
- [21] L. D. Herrnstein RJ, Rachlin H. The matching law: papers in psychology and economics. *Harvard University Press*, 1997.
- [22] K. Iigaya. Adaptive learning and decision-making under uncertainty by metaplastic synapses guided by a surprise detection system. *Elife*, 5:e18073, 2016.
- [23] K. Iigaya, M. S. Fonseca, M. Murakami, Z. F. Mainen, and P. Dayan. The long and the short of serotonergic stimulation: Optogenetic activation of dorsal raphe serotonergic neurons changes the learning rate for rewards. *bioRxiv*, page 215400, 2017.
- [24] K. Iigaya and S. Fusi. Dynamical regimes in neural network models of matching behavior. *Neural computation*, 25:1–20, 2013.
- [25] K. Iigaya, L. Sugrue, G. Corrado, W. Newsome, and S. Fusi. Deviations from the matching law reflect reward integration over multiple timescales. *Cosyne Abstract*, 2013.
- [26] H. F. Kim, A. Ghazizadeh, and O. Hikosaka. Dopamine neurons encoding long-term memory of object value for habitual behavior. *Cell*, 163(5):1165–1175, 2015.
- [27] H. F. Kim and O. Hikosaka. Distinct Basal Ganglia Circuits Controlling Behaviors Guided by Flexible and Stable Values. *Neuron*, Aug 2013.
- [28] K. P. Kording, J. B. Tenenbaum, and R. Shadmehr. The dynamics of memory as a consequence of optimal adaptation to a changing body. *Nature neuroscience*, 10(6):779–786, 2007.
- [29] G. La Camera, A. Rauch, D. Thurbon, H.-R. Lüscher, W. Senn, and S. Fusi. Multiple time scales of temporal response in pyramidal and fast spiking cortical neurons. *Journal of neurophysiology*, 96(6):3448–3464, 2006.

- [30] B. Lau and P. W. Glimcher. Dynamic response-by-response models of matching behavior in rhesus monkeys. *J Exp Anal Behav*, 84(3):555–579, 2005.
- [31] B. Lau and P. W. Glimcher. Value representations in the primate striatum during matching behavior. *Neuron*, 58(3):451–463, 2008.
- [32] Y. Loewenstein. Robustness of learning that is based on covariance-driven synaptic plasticity. *PLoS Comput. Biol.*, 4(3):e1000007, Mar 2008.
- [33] Y. Loewenstein, D. Prelec, and H. S. Seung. Operant matching as a Nash equilibrium of an intertemporal game. *Neural Comput*, 21:2755–2773, Oct 2009.
- [34] Y. Loewenstein and H. S. Seung. Operant matching is a generic outcome of synaptic plasticity based on the covariance between reward and neural activity. *Proc. Natl. Acad. Sci. U.S.A.*, 103:15224–15229, Oct 2006.
- [35] B. N. Lundstrom, A. L. Fairhall, and M. Maravall. Multiple timescale encoding of slowly varying whisker stimulus envelope in cortical and thalamic neurons in vivo. *Journal of Neuroscience*, 30(14):5071–5077, 2010.
- [36] M. R. Nassar, K. M. Rumsey, R. C. Wilson, K. Parikh, B. Heasley, and J. I. Gold. Rational regulation of learning dynamics by pupil-linked arousal systems. *Nat. Neurosci.*, 15(7):1040–1046, Jul 2012.
- [37] M. R. Nassar, R. C. Wilson, B. Heasley, and J. I. Gold. An approximately Bayesian delta-rule model explains the dynamics of belief updating in a changing environment. *J. Neurosci.*, 30(37):12366–12378, Sep 2010.
- [38] T. Neiman and Y. Loewenstein. Reinforcement learning in professional basketball players. *Nature communications*, 2:569, 2011.
- [39] T. Neiman and Y. Loewenstein. Covariance-based synaptic plasticity in an attractor network model accounts for fast adaptation in free operant learning. *Journal of Neuroscience*, 33(4):1521–1534, 2013.
- [40] Y. Niv, N. D. Daw, D. Joel, and P. Dayan. Tonic dopamine: opportunity costs and the control of response vigor. *Psychopharmacology*, 191(3):507–520, 2007.
- [41] J. M. Pearce and G. Hall. A model for pavlovian learning: variations in the effectiveness of conditioned but not of unconditioned stimuli. *Psychological review*, 87(6):532, 1980.
- [42] M. L. Platt and P. W. Glimcher. Neural correlates of decision variables in parietal cortex. *Nature*, 400:233–238, Jul 1999.
- [43] C. Pozzorini, R. Naud, S. Mensi, and W. Gerstner. Temporal whitening by power-law adaptation in neocortical neurons. *Nature neuroscience*, 16(7):942–948, 2013.
- [44] D. Prelec. Matching, maximizing, and the hyperbolic reinforcement feedback function. *Psychological Review*, 89(3):189, 1982.
- [45] K. Preuschoff and P. Bossaerts. Adding prediction risk to the theory of reward learning. *Annals of the New York Academy of Sciences*, 1104(1):135–146, 2007.

- [46] R. A. Rescorla, A. R. Wagner, et al. A theory of pavlovian conditioning: Variations in the effectiveness of reinforcement and nonreinforcement. *Classical conditioning II: Current research and theory*, 2:64–99, 1972.
- [47] A. Roxin and S. Fusi. Efficient partitioning of memory systems and its importance for memory consolidation. *PLoS Comput. Biol.*, 9(7):e1003146, Jul 2013.
- [48] R. B. Rutledge, S. C. Lazzaro, B. Lau, C. E. Myers, M. A. Gluck, and P. W. Glimcher. Dopaminergic drugs modulate learning rates and perseveration in Parkinson’s patients in a dynamic foraging task. *J. Neurosci.*, 29:15104–15114, Dec 2009.
- [49] Y. Sakai and T. Fukai. The actor-critic learning is behind the matching law: matching versus optimal behaviors. *Neural Comput*, 20(1):227–251, 2008.
- [50] Y. Sakai and T. Fukai. When does reward maximization lead to matching law? *PLoS ONE*, 3:e3795, 2008.
- [51] W. Schultz, P. Dayan, and P. R. Montague. A neural substrate of prediction and reward. *Science*, 275(5306):1593–1599, Mar 1997.
- [52] H. S. Seung. Learning in spiking neural networks by reinforcement of stochastic synaptic transmission. *Neuron*, 40:1063–1073, Dec 2003.
- [53] A. Soltani, D. Lee, and X. J. Wang. Neural mechanism for stochastic behaviour during a competitive game. *Neural Networks*, 19(8):1075 – 1090, 2006.
- [54] A. Soltani and X.-J. Wang. A Biophysically Based Neural Model of Matching Law Behavior: Melioration by Stochastic Synapses. *J. Neurosci.*, 26(14):3731–3744, 2006.
- [55] L. P. Sugrue, G. S. Corrado, and W. T. Newsome. Matching Behavior and the Representation of Value in the Parietal Cortex. *Science*, 304(5678):1782–1787, 2004.
- [56] R. S. Sutton. Td models: Modeling the world at a mixture of time scales. In *ICML*, volume 12, pages 531–539. Citeseer, 1995.
- [57] R. S. Sutton and A. G. Barto. *Reinforcement Learning: An Introduction (Adaptive Computation and Machine Learning)*. A Bradford Book, Mar. 1998.
- [58] N. Ulanovsky, L. Las, D. Farkas, and I. Nelken. Multiple time scales of adaptation in auditory cortex neurons. *Journal of Neuroscience*, 24(46):10440–10453, 2004.
- [59] I. Vilares, J. D. Howard, H. L. Fernandes, J. A. Gottfried, and K. P. Kording. Differential representations of prior and likelihood uncertainty in the human brain. *Current Biology*, 22(18):1641–1648, 2012.
- [60] X. J. Wang. Probabilistic decision making by slow reverberation in cortical circuits. *Neuron*, 36(5):955–968, Dec 2002.
- [61] B. Wark, A. Fairhall, and F. Rieke. Timescales of inference in visual adaptation. *Neuron*, 61(5):750–761, 2009.
- [62] C. J. Watkins and P. Dayan. Q-learning. *Machine learning*, 8(3-4):279–292, 1992.

- [63] C. J. C. H. Watkins. *Learning from delayed rewards*. PhD thesis, King's College, Cambridge, 1989.
- [64] R. C. Wilson, M. R. Nassar, and J. I. Gold. A mixture of delta-rules approximation to bayesian inference in change-point problems. *PLoS Comput. Biol.*, 9(7):e1003150, Jul 2013.
- [65] M. K. Wittmann, N. Kolling, R. Akaishi, B. K. Chau, J. W. Brown, N. Nelissen, and M. F. Rushworth. Predictive decision making driven by multiple time-linked reward representations in the anterior cingulate cortex. *Nature Communications*, 7, 2016.

Reviewers' Comments:

Reviewer #2:

Remarks to the Author:

I very much appreciate the authors' careful and thoughtful replies, as well as their additional clarifications. I am happy to endorse publication.